# Winning the Lottery Once and For All: Towards Pruning Neural Networks at Initialization

**David S. Hippocampus**[*]
**Department of Computer Science**
**Cranberry-Lemon University**
**Pittsburgh, PA 15213**
`hippo@cs.cranberry-lemon.edu`

## Abstract

Lottery Ticket Hypothesis (LTH) posits the existence of *winning tickets*, i.e., sparse subnetworks within randomly initialized dense neural networks that are capable of achieving test accuracy comparable to the original, unpruned counterpart when trained from scratch, with an optimal learning rate and in a similar training budget. Despite this promising conjecture, recent studies (Liu et al., 2018; Frankle et al., 2020a; Su et al., 2020; Ma et al., 2021a) have cast doubt on the feasibility of identifying such winning tickets at initialization, particularly in large-scale settings. They suggest that in such expansive environments, winning tickets exclusively and only emerge during the early phase of training. This observation, according to Frankle and Carbin (2018), contradicts the core tenet of LTH as these winning tickets do not truly *win the initialization lottery*. In light of recent findings, we address a critical question: **If winning tickets can only be obtained during early iterations, does the initial training phase of a neural network encode vital knowledge, which we refer to as lottery-ticket information, that can be utilized to generate winning tickets at initialization, especially in large-scale scenarios?**

We affirmatively answer this question by introducing a novel premise, Knowledge Distillation-based Lottery Ticket Search (KD-LTS). Our framework harnesses latent response, feature, and relation-based lottery-ticket information from an ensemble of teacher networks, employing a series of deterministic approximations to address an intractable Mixed Integer Optimization problem. This enables us to consistently *win the initialization lottery* in complex settings, identifying winning tickets right from the initialization point at unprecedented sparsity levels - achieving as high as 95% for VGG-16 and 65% for ResNet-20, and accomplishing this 19 times faster than Iterative Magnitude Pruning (IMP). Remarkably, without bells and whistles, even winning tickets identified early in the training process using our technique - consistently yield a performance gain of 2% for VGG-16 and 1.5% for ResNet-20 across various levels of sparsity, thereby surpassing existing methods. Furthermore, our work is the first in the literature to obtain state-of-the-art results in winning tickets at extreme sparsity levels across diverse experimental setups, all while successfully passing the comprehensive sanity checks referenced in prior work (Ma et al., 2021a; Frankle et al., 2020a; Su et al., 2020), thereby reinstating the possibility for pruning at initialization.

## 1 Introduction

In the ever-evolving landscape of machine learning, the demand for integrating the intelligence of Deep Neural Networks (DNNs) into diverse applications has surged. Recent breakthroughs in DNNs have facilitated remarkable advancements in challenging domains such as, computer vision (Krizhevsky et al., 2012; Taigman

---

[*]Use footnote for providing further information about author (webpage, alternative address)—*not* for acknowledging funding agencies.

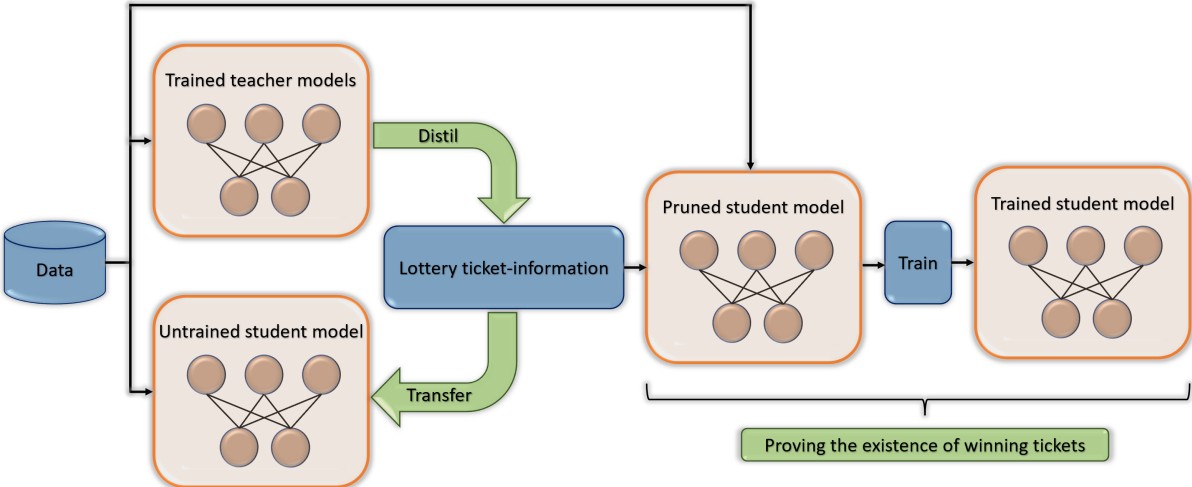

Figure 1: Illustration of our Knowledge Distillation inspired Lottery Ticket Search (KD-LTS).

et al., 2014; Xiao et al., 2018) - enabling feats such as real-time facial recognition and autonomous vehicle navigation; reinforcement learning (Silver et al., 2016; Mnih et al., 2015) - mastering complex games and outperforming human experts; speech recognition (Graves et al., 2013; Hannun et al., 2014) - bringing us virtual assistants; and in machine translation (Sutskever et al., 2014) - breaking language barriers, among others. However, the pursuit of phenomenal performance has often come at a steep price to pay: excessive overparameterization (Simonyan and Zisserman, 2014). This tendency towards deploying increasingly larger and more complex networks, while sometimes necessary for achieving top-tier results, has raised legitimate concerns. Consider the immense computational and storage costs - often needing billions of parameters; the environmental impact - training models like BERT can have a carbon footprint comparable to a trans America flight for one person; and the financial cost - training advanced models such as XLNet is estimated to be around US $61,000 (Erion et al., 2022; Verdecchia et al., 2023; Heikkilä, 2022; Peng and Sarazen, 2019). This issue has impeded practical deployment of DNNs in resource-constrained real-world environments, such as on-device inference in mobile phones, and similar edge devices (Zhu and Gupta, 2017).

**Post vs pre-training pruning.** To tackle this challenge, network pruning has emerged as a promising technique (Han et al., 2015; Hu et al., 2016; Mariet and Sra, 2015; Dong et al., 2017; Yang et al., 2017; Louizos et al., 2017), allowing DNNs to be significantly pruned during inference without sacrificing test accuracy. For instance, pruning techniques have achieved a 1.5x decrease in parameter count for Convolutional Neural Networks Li et al. (2016), alongside a notable 20.79% cut in inference time for ResNet-50, now just 17.22 milliseconds (Gong et al., 2022), all while maintaining a minimal drop in accuracy, often less than 1%. Expanding the application of pruning beyond mere inference efficiency improvements, researchers are additionally exploring early pruning in neural networks to reduce training costs (Strubell et al., 2019). However, this strategy can result in less trainable architectures with decreased performance compared to the original network, suggesting that a certain degree of overparameterization might still be necessary for successful training (Livni et al., 2014).

**Pruning at initialization: missing the mark? Towards early-stage pruning.** Contrary to conventional notion that sparse networks can only be obtained through post-training pruning, LTH (Frankle and Carbin, 2018) postulated a paradigm-shift by unveiling high-performing, sparse subnetworks - called winning tickets - right at initialization. Employing the Iterative Magnitude Pruning (IMP) (Han et al., 2015) technique with resetting mechanism, Frankle and Carbin (2018) successfully uncovered these winning tickets in Conv-2, Conv-4, and Conv-6 architectures trained on MNIST and CIFAR-10, achieving high levels of sparsity (80-90% or more). However, these promising results were predominantly limited to simpler settings. The practical success of LTH has been profound, demonstrating wide applicability across various domains (Evci et al., 2022; Yu et al., 2019; Kalibhat et al., 2021) and inspiring a multitude of follow-up studies to identify winning tickets at initialization, (Lee et al., 2018; de Jorge et al., 2020; Wang et al., 2020; Sreenivasan et al., 2022;

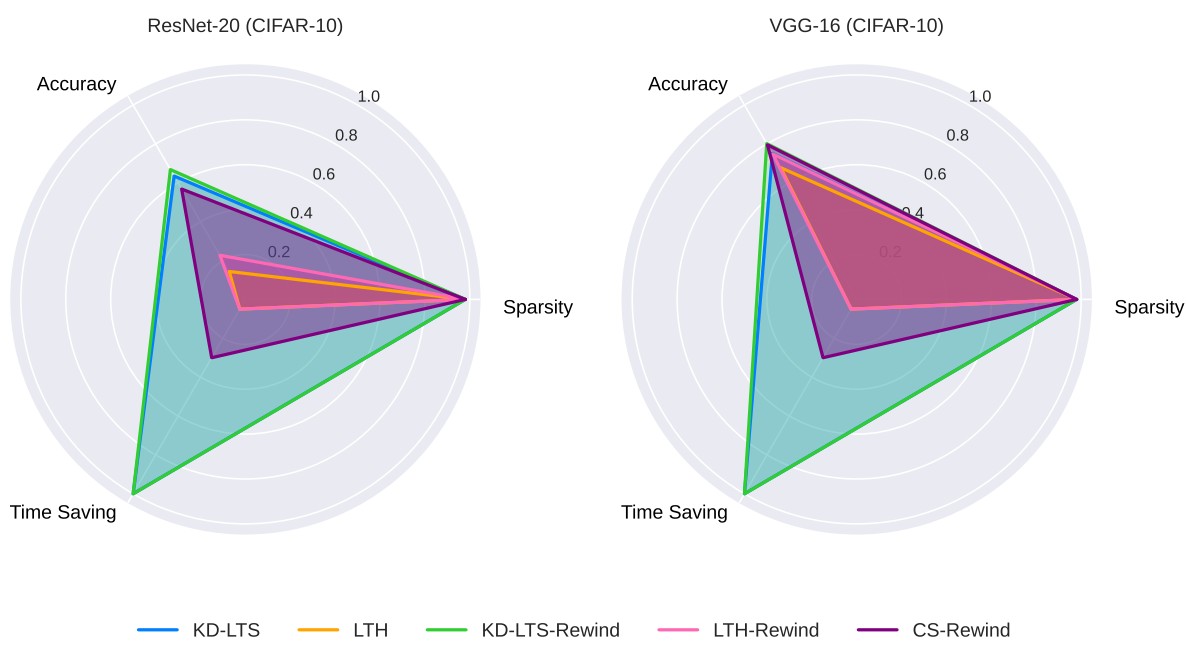

Figure 2: Comparative evaluation of KD-LTS' winning ticket performance in VGG-16 and ResNet-20 architectures on the CIFAR-10 dataset, extracted at initialization and early epoch contrasted with LTH, LTH-Rewind, and CS. Note: Scale of the accuracy axis has been adjusted to $[0.7, 1]$.

Alizadeh et al., 2022; Tanaka et al., 2020). Despite practical success, recent studies (Gale et al., 2019; Frankle et al., 2020c; Savarese et al., 2019; Frankle et al., 2020a) highlight the limitations of these pruning techniques, including IMP, in more demanding large-scale settings. Specifically, for complex architectures like VGGs and ResNets trained on CIFAR-10/CIFAR-100 and ImageNet, these methods struggle to identify winning tickets with impactful sparsity levels. As illustrated in Figure 3, these methods only manage to extract winning tickets at trivial sparsities - $\sim 20\%$ or even less - in larger networks such as VGG-16, ResNet-18, and ResNet-20, marking a stark contrast to the performance seen in smaller-scale scenarios.

Building on the critiques outlined by recent research, investigations (Su et al., 2020; Frankle et al., 2020b; Sreenivasan et al., 2022; Ma et al., 2021a; Frankle et al., 2020a) into existing pruning-at-initialization techniques have further underscored their limitations through baseline tests that incorporate both sensitivity and stability analyses. These studies demonstrate that randomly sampled sparse subnetworks, when adjusted for carefully chosen layer-wise sparsity, frequently outperform those identified by prevailing pruning-at-initialization methods. Furthermore, these subnetworks exhibit significant vulnerability to perturbations introduced by Stochastic Gradient Descent (SGD) noise, shedding light on their comparative under-performance. For an in-depth examination and insights into sensitivity and stability analyses, readers are encouraged to refer to Sections 6 and 6, respectively.

In addressing the challenges highlighted by these assessments, Frankle et al. (2020a) explored modifications to the 'IMP with resetting technique' to better suit large-scale settings. Their modified approach successfully identified winning tickets while passing both stability and sensitivity tests. However, this advancement compromised the core premise of the LTH: the identification of winning tickets strictly at the network's initialization. Their revised methodology requires pre-training the dense network for a few initial epochs, followed by rewinding the weights to these early epochs rather than resetting them to the original initialization point. This departure from pruning-at-initialization necessitates substantial hardware and software resources for pre-training the dense network for at least a few initial epochs.

Given the paramount importance of identifying winning tickets at initialization to significantly simplify training costs for practical real-world deployments, the current state of affairs regarding pruning at initialization in large-scale settings brings us to a critical juncture: **Do winning tickets with non-trivial sparsity exist**

**at initialization in complex settings? Moreover, can these tickets not only demonstrate tangible accuracy advantages compared to the original, dense networks but also meet the standards of stability and sensitivity analyses to ascertain their credibility?** To date, to the best of our knowledge, these two questions remain open problems.

**Knowledge Distillation-based Lottery Ticket Search: Towards pruning at initialization.** In this study, we delve into these open questions by establishing a novel link between the valuable information encoded within neural networks during their various training phases and the identification of winning tickets at initialization - a connection that has not been explored before.

---

**Key Insight**

Knowledge Distillation-aided Lottery Ticket Search (KD-LTS) efficiently distills a rich tapestry of critical response, feature, and relation-based heterogeneous information - referred to as lottery-ticket information - from an ensemble of trained teacher networks, in addition to leveraging the information contained in data labels, to identify subnetworks within a dense randomly initialized student network. This pioneering approach facilitates the task of identifying winning tickets - a feat that existing pruning-at-initialization techniques, which rely on traditional model training solely with labels, could not accomplish.

---

In brief, the KD-LTS framework is structured around three key components. **Phase-1** involves precise identification of diverse lottery-ticket information across an array of trained teacher networks. In **Phase-2**, KD-LTS introduces a series of deterministic relaxations to a Mixed Integer Optimization problem, a crucial step to effectively train binary masks within a randomly initialized dense student network. This step facilitates the efficient transfer of identified lottery-ticket information from the teacher models. The final component, **Phase-3**, incorporates a unique custom-designed scoring function, wrapped around a Straight-Through Estimator (STE), a freezing mechanism, and a regularization term. This combination effectively orchestrates control of sparsity, playing a pivotal role in striking an optimal balance between sparsity and performance. Ultimately, these three components synergize to equip subnetworks at initialization with the essential lottery-ticket information, thus qualifying them as winning tickets. Detailed information about the KD-LTS methodology can be found in Section 4, and an illustrative representation of this framework is provided in Figure 14. The complete code is publicly available at `https://github.com/gap48/Winning-the-lottery`.

**Contributions.** We summarize our contributions as follows:

(*i*) **Revitalizing pruning-at-initialization:** We introduce KD-LTS, a novel methodology that harnesses knowledge distillation to resurrect the potential for pruning at initialization. As highlighted in Figure 2, KD-LTS outperform state-of-the-art techniques, demonstrating Pareto-optimal performance in terms of sparsity, accuracy, and computational efficiency. Through extensive experiments and comprehensive evaluation of training settings and quality analyses, we formalize a strict definition of LTH in Section 6.3 to standardize and catalyze research on pruning at initialization, and rigorously validate KD-LTS winning tickets' compliance with this definition, thus ensuring their reliability and effectiveness.

(*ii*) **Empirical validation above and beyond overall performance:** KD-LTS framework facilitates the identification of subnetworks that exhibit significant improvements over existing benchmarks, including significant gains in test accuracy ($\uparrow 2\%$), sparsity ($\uparrow 40\%$) of winning tickets at initialization, test accuracy ($\uparrow 1\%$), sparsity ($\uparrow 5\%$) of winning tickets at early epoch, and supermasks ($\uparrow 3\%$). Additionally, we observed nontrivial enhancements in mCE, ROC-AUC, adversarial robustness, ECE, NLL, Stability, Fidelity, and Transferability ($\uparrow 1$-$5\%$), thereby meeting the constantly advancing and contrasting demands in applications of DNNs. Please refer to Sections 5, C, and D.1 for more details. Furthermore, we uncover a surprising and non-intuitive interplay between specialized characteristics (e.g., adversarial robustness) of teacher models and the properties of extracted winning tickets, as presented in Section A. This suggests the potential for strategically embedding such traits beyond mere improvement in performance regarding that trait.

(*iii*) **Practical utility of KD-LTS:** Advancing beyond the utility of KD-LTS as merely a research exploration into the dynamics of pruning at initialization, this study broadens the scope, showcasing its effectiveness

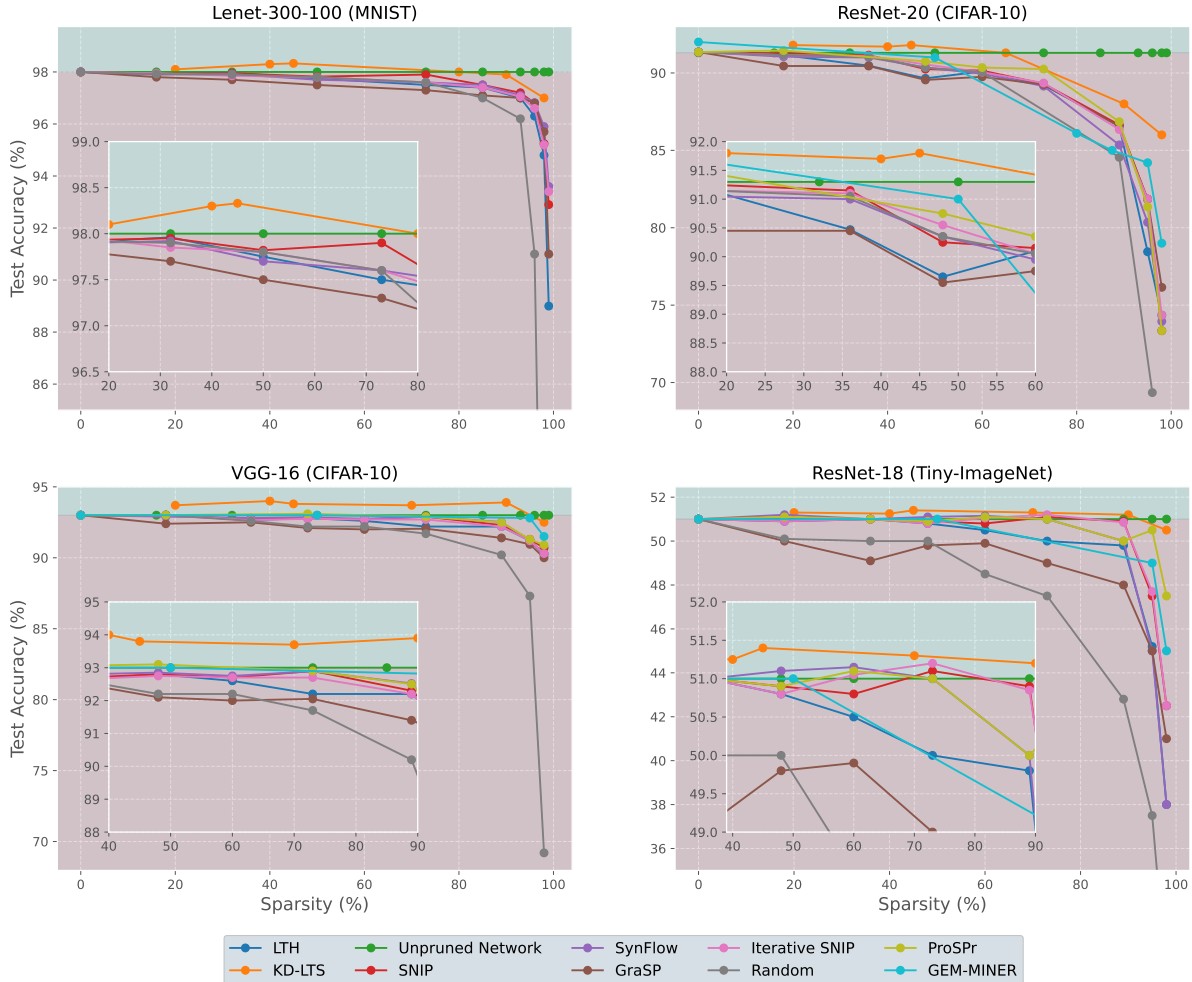

Figure 3: Comparative assessment of KD-LTS' winning ticket performance across diverse networks and datasets, juxtaposed against leading methods for pruning at initialization, including LTH, SNIP, Iterative SNIP, GraSP, SynFlow, ProsPr, and GEM-MINER.

as a practical tool for the extraction of winning tickets. Comprehensive ablation studies on the key components of KD-LTS, as illustrated in Section B, illuminate the technique's capacity to identify winning tickets with significant sparsity by utilizing just a single teacher model trained for merely three epochs, using only 70% of the training time and 70% of the full dataset, and leveraging a bare minimum response-based knowledge category.

(*iv*) **Transforming knowledge distillation with KD-LTS:** KD-LTS not only challenges conventional approaches to pre-training pruning but also knowledge distillation, offering innovative new perspectives within both fields. Diverging from traditional knowledge distillation methods that transfer knowledge from large teacher models to smaller students through weight training, KD-LTS leverages large student networks and employs a novel mask-training technique for knowledge transfer. As highlighted in Section D.1, student networks pruned using KD-LTS demonstrate significant performance improvements and a substantial reduction in parameter counts compared to small, dense student networks trained through standard knowledge transfer methods.

## 2 Related Works

**Pruning after initialization.** Studies (Zhang et al., 2021a; Du et al., 2018; Neyshabur et al., 2018; Allen-Zhu et al., 2019) have underscored the requisite of overparameterization in the performance of DNNs. To tackle the challenges associated with overparameterization, a range of model compression techniques have been developed. These approaches include weight factorization (Denton et al., 2014), weight sharing (Wu et al., 2018), quantization (Hubara et al., 2017), network architecture search (Zoph and Le, 2016), dynamic inference (Wang et al., 2018), and notably, pruning (Han et al., 2015) and knowledge distillation (Hinton et al., 2015). Conventional pruning methods typically employ saliency criteria such as weight magnitude (Han et al., 2015), gradients (Molchanov et al., 2016; 2019), Taylor expansion (Molchanov et al., 2016), movement analysis (Sanh et al., 2020), and second-order derivatives (LeCun et al., 1990) for the selective removal of weights in neural networks. This study represents a notable shift from traditional methods, transcending beyond the limitations of ad-hoc criteria and fixed pruning intervals typically associated with conventional approaches. We implement a series of deterministic optimization problems within a framework that enables the continuous re-calibration of pruned weights. This methodology not only facilitates the potential recovery of pruned weights but also offers a more dynamic and adaptive approach to network sparsification.

**Knowledge distillation.** Alongside pruning, knowledge distillation has emerged as a potent strategy to counter overparameterization, offering significant advancements in model compression since its initial conceptualization by Ba and Caruana (2014) and Hinton et al. (2015). Knowledge categories have been extensively studied, with notable strategies including: • **Response-based** soft target Knowledge Distillation (KD) (Hinton et al., 2015) • **Feature-based** Mean Squared Error (MSE) loss (Ba and Caruana, 2014), and first-order Jacobian information (Czarnecki et al., 2017) • **Relational-based** attention maps (Zagoruyko and Komodakis, 2016), Contrastive Representation Distillation (CRD) framework (Tian et al., 2019), critical period-aware mutual information between features (PKT) Passalis et al. (2020), (RKD) Park et al. (2019), to leverage sample and feature diversity, the solution of the flow procedure (FSP) (Yim et al., 2017) to exploit layer diversity, and Adaptive Ensemble Knowledge Distillation (AE-KD) (Malinin et al., 2019) to utilize diversity within ensembles. We leverage this rich history of research and use the aforementioned knowledge categories in our experiments. The selection of knowledge types plays a crucial role in shaping the learning process of the student model. Consequently, the objective is to systematically identify key latent information imprinted within a trained teacher model and carefully curate them according to specific applications. The applications of knowledge distillation span diverse domains such as computer vision (Kim and Rush, 2016; Papernot et al., 2016; Yu et al., 2017), sequence modeling (Kim and Rush, 2016), semi-supervised learning (Tarvainen and Valpola, 2017), hyperparameter search (Liu et al., 2023), training efficiency improvement(Blakeney et al., 2022), and multi-modal learning (Gupta et al., 2016), among others. KD-LTS extends the applications of knowledge distillation to an unexplored domain - pruning at initialization.

**Overall assessment of winning tickets.** LTH impacted research on pruning at initialization, leading to the development of techniques such as SNIP (Lee et al., 2018), GraSP (Wang et al., 2020), SynFlow (Tanaka et al., 2020), Iterative SNIP (Verdenius et al., 2020; de Jorge et al., 2020), ProsPr (Alizadeh et al., 2022), and GEM-MINER (Sreenivasan et al., 2022). This milieu of studies on extracting subnetworks has fostered the practical applicability of LTH in image classification (Evci et al., 2022; You et al., 2019; Savarese et al., 2019), object detection (Girish et al., 2021), natural language processing (Yu et al., 2019; Prasanna et al., 2020; Chen et al., 2021a), generative adversarial networks (Kalibhat et al., 2021), graph neural networks (Chen et al., 2021a), reinforcement Learning (Yu et al., 2019), lifelong learning (Chen et al., 2020), and the introduction of supermasks (Zhou et al., 2019). In addition, this has stimulated the assessment of subnetworks in various contexts, such as transferability across datasets, optimizers, and architectures (Chen et al., 2021b; Morcos et al., 2019), enhancing fairness (Hooker et al., 2019; Paganini, 2020), improving robustness (Gui et al., 2019; Ye et al., 2019; Chen et al., 2022), addressing privacy concerns (Wang et al., 2019), and providing tools for uncertainty estimation (Venkatesh et al., 2020). Nonetheless, the majority of evaluations have predominantly focused on tickets derived from the early training phase. Our work extends the horizon of LTH by identifying supermasks that surpass existing methods (Sreenivasan et al., 2022; Ramanujan et al., 2020; Savarese et al., 2019; Zhou et al., 2019), and showcasing that KD-LTS' winning tickets, identified at initialization, can meet or even exceed the benchmarks across a wide range of assessments typically associated with post-initialization pruning or dense network training.

Table 1: Overview of key attributes in initialization and early phase pruning methods, including winning the jackpot (securing winning tickets at non-trivial sparsity levels).

| Pruning method | Stability | Sensitivity | Prunes at initialization | Wins the jackpot | Cost |
|:---:|:---:|:---:|:---:|:---:|:---:|
| IMP | ✗ | ✗ | ✓ | ✗ | 2850 epochs |
| SNIP | ✗ | ✗ | ✓ | ✗ | 1 epoch |
| GraSP | ✗ | ✗ | ✓ | ✗ | 1 epoch |
| LTH-Rewind | ✓ | ✓ | ✗ | ✓ | 3000 epochs |
| Synflow | ✗ | ✗ | ✓ | ✗ | 3 epochs |
| ProsPr | ✗ | ✗ | ✓ | ✗ | 200 epochs |
| Gem-Miner | ✗ | ✓ | ✓ | ✗ | 220 epochs |
| CS | ✓ | ✓ | ✗ | ✓ | 360 epochs |
| KD-LTS | ✓ | ✓ | ✓ | ✓ | 150 epochs |

## 3  Notation

We consider a $C$-class classification problem, where $\mathcal{X} \subseteq \mathbb{R}^d$ represents the feature space and $\mathbb{D} = \{(x_i, y_i)\}_{i=1}^I$ is the dataset. Every sample $x_i$ is associated with a class $c$ drawn from set $\{1, 2, \ldots, C\}$ and $y_i$ is a one-hot vector which has a value of 1 in the $c$-th component to represent ground-truth label $c$, and a value of 0 in all other components. Our objective is to identify winning tickets in a neural network that models the mapping of input features to corresponding labels, denoted as $f_L^s(x; m \odot \theta) : \mathcal{X} \to \mathbb{R}^C$. The parameters of this network are represented as $\theta \in \mathbb{R}^P$. Here, 's' represents the student model, $L$ represents the penultimate layer in the set of layers $\{1, 2, \ldots, L\}$, $m$ is a trainable binary mask, i.e., $m \in \{0, 1\}^P$ and $\odot$ refers to element-wise multiplication. We denote a vector of all-ones as $\mathbf{1}$. In the context of a dense network, where $m = \mathbf{1}$, the network is represented as $f_L^s(x; \theta)$. We refer to a teacher model as $f_L^{t_k}$, where $k \in \{1, 2, \ldots, K\}$ represents the index of the teacher model in the ensemble. To identify a specific element $i$ in the feature map associated with layer $l$ of the neural network, we use the notation $f_{il}$. The training process spans a total of $E$ epochs. For each epoch $e \in \{1, 2, \ldots, E\}$ the associated parameters are denoted as $\theta^{(e)}$, and the corresponding binary masks are represented as $m^{(e)}$. Sparsity $S$ of the pruned network is calculated as $S = 1 - \frac{\|m\|_0}{P}$.

## 4  Knowledge Distillation-based Lottery Ticket Search (KD-LTS)

**Definition 4.1** (Subnetwork)**.** For a dense network $f_L(x; \theta) : \mathcal{X} \to \mathbb{R}^C$, a subnetwork is denoted as $f_L(x; m \odot \theta)$ and is defined by a binary mask $m \in \{0, 1\}^P$. Here, a parameter component $\theta_u$ is retained if $m_u = 1$ and removed otherwise. For any given configuration of $m$, the effective parameter space of the resultant subnetwork is characterized by $\|m\|_0$ parameters.

A key limitation of current early-pruning practices is the necessity to train the network before it can be pruned. The subsequent subsection elaborates on the multifaceted components of the KD-LTS framework, which combines response-based, feature-based, and relational-based lottery-ticket information in addition to learning from one-hot hard labels as output targets, allowing it to overcome this limitation. Harnessing this heterogeneous information from trained teacher models, KD-LTS crafts a subnetwork within the student network that mirrors the essential characteristics of the trained teacher model - the very attributes that make this trained phase of the teacher model contain winning tickets, a phase that prevailing early-pruning strategies specifically capitalize on.

### 4.1  First Stop: Distilling Knowledge

**Learning from ground-truth label.** The student network $f_L(x, (m \odot \theta))$ outputs a probabilistic class posterior $\mathcal{P}(z|x; (m \odot \theta))$ for a given sample $x$ over class $c$, which is formulated as follows:

$$\mathcal{P}(z|x; (m \odot \theta)) = \text{Cat}\left(\frac{e^{f_L(x;\theta)}}{\sum_{c=1}^C e^{f_{cL}(x;\theta)}}\right).$$

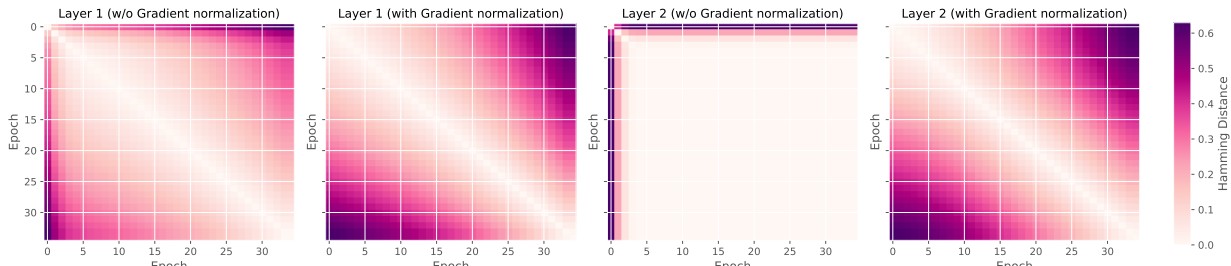

Figure 4: Hamming distance of masks across epochs within a KD-LTS framework for a LeNet-300-100/MNIST experiment, with and without gradient normalization.

In this expression, $z$ represents the logits or unnormalized log probabilities generated by the network, $e^{(\cdot)}$ denotes element-wise exponentiation and 'Cat' signifies Categorical distribution. The conventional objective function for training a multi-class classification model is the cross-entropy measurement between the predicted and ground-truth label distributions, defined as:

$$L_{\mathrm{CE}}(x; (m \odot \theta)) = \frac{1}{I} \sum_{i=1}^{I} y_i^{\mathsf{T}} \log \mathcal{P}(z_i | x_i; (m \odot \theta)).$$

Here, $\mathsf{T}$ denotes the transpose operator. $L_{\mathrm{CE}}$ guides the network towards predicting the correct class label through maximum likelihood estimation.

**Response-based distillation.** Our response-based distillation aligns the logits of the student network with those of the teacher network, softened by a high temperature $T$. This softening, achieved by higher values of $T$, results in smoother distributions. This distillation loss function, accounting for the network response, is expressed as:

$$L_{\mathrm{KD}}(x; (m \odot \theta)) = \frac{1}{IK} \sum_{i=1}^{I} \sum_{k=1}^{K} \nu_k \left( \frac{e^{\frac{f_L^{\mathrm{t}_k}(x_i; \theta)}{T}}}{\sum_{c=1}^{C} e^{\frac{f_{cL}^{\mathrm{t}_k}(x_i; \theta)}{T}}} \right)^{\mathsf{T}} \log \mathcal{P}(z_i | x_i; (m \odot \theta)).$$

In this formulation, $\nu_k$ is the weight assigned to the $k^{th}$ teacher, and $\log(\cdot)$ is applied element-wise. $L_{\mathrm{KD}}$ minimizes the Kullback-Leibler (KL) divergence to effectively transfer nuanced, class-discriminative information from the teacher to the student.

**Feature-based distillation.** Inspired by Czarnecki et al. (2017), our feature-based distillation loss for aligning internal representations goes beyond conventional considerations of target information by incorporating Jacobian information alongside target information. In the pursuit of extracting winning tickets, the feature-based loss, denoted as $L_{\mathrm{FD}}$ utilizing Sobolev training, is formulated as follows:

$$L_{\mathrm{FD}} = \frac{1}{IK} \sum_{i=1}^{I} \sum_{k=1}^{K} \nu_k \left\| \frac{\nabla_\theta f_L^{\mathrm{t}_k}(x_i; \theta)}{\left\| \nabla_\theta f_L^{\mathrm{t}_k}(x_i; \theta) \right\|_2} - \frac{\nabla_{\theta^{(0)}} f_L^{\mathrm{s}}(x_i; (m \odot \theta^{(0)}))}{\left\| \nabla_{\theta^{(0)}} f_L^{\mathrm{s}}(x_i; (m \odot \theta^{(0)})) \right\|_2} \right\|_2^2.$$

Here, $\nabla_\theta f_L^{\mathrm{t}}(x; \theta)$ and $\nabla_{\theta^{(0)}} f_L^{\mathrm{s}}(x; (m \odot \theta^{(0)}))$ represent the first-order Jacobian matrices of the final layer outputs of the teacher and student networks with respect to their parameters, respectively. By employing the Sobolev loss function, we encourage the student network's parameters to replicate the gradient flow of the trained teacher network. As a result, the parameters of the subnetwork identified within the student model emulate a learning pattern similar to that of the teacher network, thereby fostering a stable learning platform post-extraction.

**Algorithm 1:** KD-LTS

**Initialization**: Initialize $s^{(0)}$, penalty $\lambda$, iterations $T$, $\alpha$, $\gamma$, parameter $\theta^{(0)}$, multiplication factor $\mu$, $\hat{c}$

**for** epoch $e$ in $E$ **do**

    **for** iteration $n$ in $N$ **do**

        Solve for $L_M$:

$$\min_{s^{(e)}} L_M\left(x; r\left(\sigma\left(\beta^{(e)}s\right)\right)\odot\theta^{(0)}\odot q\right) + \lambda\left\|\sigma\left(\beta^{(e)}s\right)\right\|_1.$$

        /*** STE ***/

        $p_{\text{sorted}} \leftarrow \text{Sort}\left(\sigma\left(\beta^{(e)}s\right)\right).$

        $p_{\text{bottom}} \leftarrow \text{Bottom-}(1-e^{-\hat{c}e})$ fraction of $p_{\text{sorted}}$.

        $r \leftarrow r \odot 1_{p_i \notin p_{\text{bottom}}}.$

        /*** STE ***/

    **end**

    $\beta_l^{(e)} \leftarrow \mu_l\beta_l^{(e)}.$

    **if** $\text{mod}(e, V) = 0$ **then**

        $I_2 \leftarrow \{i : q_i = 1\}.$

        $w_{\text{sorted}} \leftarrow \text{Sort}\left(\sigma\left(\left(\beta^{(e)}s\right)\right) \in I_2\right).$

        $w_{\text{bottom}} \leftarrow \text{Bottom-}(1-e^{-\hat{c}e})$ fraction of $w_{\text{sorted}}$.

        $q \leftarrow r \odot 1_{q_i \notin w_{\text{bottom}}}.$

    **end**

**end**

Fix the masks and output $f_L^s(x; (H(\beta^{(E)}s^{(E)})\odot\theta^{(0)})).$

Solve $\min_{\theta} L_{CE}\left(x; (H(\beta^{(E)}s^{(E)})\odot\theta)\right)$ to prove winning tickets.

---

**Relational-based distillation.** Drawing upon and significantly adding to the work by Yim et al. (2017) on transferring layer diversity, our relational-based loss, denoted as $L_{RD}$, captures and aligns the diversity in feature representations between teacher and student networks. This relational-based loss is written as:

$$L_{RD}(x; (m\odot\theta)) = \frac{1}{IK}\sum_{i=1}^{I}\sum_{k=1}^{K}\sum_{l=1}^{L}\nu_k D\left(\mathcal{P}_{(i,j)}^{(t,l)}, \mathcal{P}_{(i,j)}^{(s,l)}\right).$$

where the probability distributions $\mathcal{P}_{(i,j)}^{(t,l)}$ and $\mathcal{P}_{(i,j)}^{(s,l)}$ are defined as follows:

$$\mathcal{P}_{(i,j)}^{(t,l)} = \frac{\text{Ker}(f_{i(l-1)}^t, f_{jl}^t)}{\sum_{i',j',i'\neq j'}\text{Ker}(f_{i'(l-1)}^t, f_{j'l}^t)}, \quad \mathcal{P}_{(i,j)}^{(s,l)} = \frac{\text{Ker}(f_{i(l-1)}^s, f_{jl}^s)}{\sum_{i',j',i'\neq j'}\text{Ker}(f_{i'(l-1)}^s, f_{j'l}^s)}.$$

These equations define the pairwise similarity probabilities for features between layers $l-1$ and $l$ within the teacher and student networks, normalized by the sum of all pairwise kernel similarities at the same layer, excluding self-similarity (where $i'\neq j'$), thereby capturing the flow of information within each network. The choice of kernel function $\text{Ker}(\cdot)$, which evaluates the similarity between feature vectors, is crucial in this computation. We systematically experiment and adopt an angle-wise metric: $\text{Ker}(a,b) = \frac{1}{2}\left(\frac{a^\top b}{\|a\|_2\|b\|_2}+1\right)$ aimed at measuring cosine similarity to penalize structural differences in feature relations and enhance the quality of transfer. The divergence $D(\cdot,\cdot)$ between the teacher's and student's relational distributions is evaluated using the Jeffreys divergence, a symmetric version of the KL divergence, to ensure a balanced knowledge transfer process:

$$L_{RD}(x; (m\odot\theta)) = \frac{1}{KI(I-1)}\sum_{i,j,i\neq j}\sum_{k=1}^{K}\sum_{l=1}^{L}\nu_k\left(\mathcal{P}_{(i,j)}^{(t_k,l)} - \mathcal{P}_{(i,j)}^{(s,l)}\right)\left(\log\mathcal{P}_{(i,j)}^{(t_k,l)} - \log\mathcal{P}_{(i,j)}^{(s,l)}\right)$$

Through this approach, we facilitate the transfer of the teacher's efficient learning patterns onto the student network. As a result, the final holistic optimization objective of our knowledge distillation framework is to minimize the following loss function:

$$L_{\mathrm{M}}(x; (m \odot \theta)) = \alpha L_{\mathrm{CE}}(x; (m \odot \theta)) + T^2(1 - \alpha)L_{\mathrm{KD}}(x; (m \odot \theta)) + \gamma L_{\mathrm{FD}}(x; (m \odot \theta)) + \gamma L_{\mathrm{RD}}(x; (m \odot \theta)).$$

The weight coefficients, $\alpha$ and $(1 - \alpha)$, are used for seamless integration of the cross-entropy loss $L_{\mathrm{CE}}$ and the response-based knowledge distillation loss $L_{\mathrm{KD}}$, respectively. Since gradient magnitudes from soft targets $e^{\frac{f_L(x;\theta)}{T}}$ are scaled by $1/T^2$ during knowledge transfer, $L_{\mathrm{KD}}$ is multiplied by a factor of $T^2$ to ensure balanced contributions from $L_{\mathrm{CE}}$ and $L_{\mathrm{KD}}$ during training.

**Supervision scheme for different facets of distillation.** KD-LTS employs a dynamic weighting scheme for the hyperparameter $\gamma$. This scheme prioritizes aligning the intermediate layer information flow during the initial training phase, gradually shifting focus towards the task-specific knowledge in the teacher's final layer as training progresses. The weight $\gamma$ is calculated as: $\gamma^{(e)} = \gamma_{\mathrm{init}}\omega^e$ where $e$ is the current training epoch, $\omega$ is a decay factor and $\gamma_{\mathrm{init}}$ is the initial weight. Following Passalis et al. (2020), this tactic helps the student network learn essential internal representations early on, before information plasticity diminishes, thus facilitating the minimization of information flow divergence and the effective use of target labels.

## 4.2   Second Stop: Picking the Winning Ticket

Expanding on the knowledge distillation phase, the goal in this second step is to dynamically sparsify the randomly initialized student network to assimilate the essence of lottery-ticket information derived from the ensemble of trained teacher models. To achieve this, we formulate and solve the following Mixed-Integer Optimization (MIO) problem:

$$\min \quad L_{\mathrm{M}}(x; m \odot \theta^{(0)}) + \lambda \|m\|_0$$
$$\text{subject to} \quad m \in \{0,1\}^P$$

where $\lambda \geq 0$ is the regularization coefficient.

**Continuous approximation for tractability.** Similar to Savarese et al. (2019), we tackle the tractability issue of MIO by re-parameterizing the mask $m$ as a function of a new variable $s \in \mathcal{R}^P$, and introducing an approximation using a series of continuous sigmoid functions $\sigma(s) = \frac{1}{1+e^{-s}}$ indexed by $\beta \in [1, \infty)$. This leads to the reparameterization $m = \sigma(\beta s)$. Specifically, for $\beta = 1, \sigma(\beta s) = \sigma(s)$ and for $\lim_{\beta \to \infty} \sigma(\beta s) = H(s)$, where H(s) is the Heaviside function. Such an approximation approach facilitates a sequence of continuous optimization problems, thus enabling effective learning of masks. The objective for each value of $\beta$ is given by:

$$L_{\mathrm{M}}\left(x; \sigma\left(\beta s\right) \odot \theta^{(0)}\right) + \lambda \|\sigma\left(\beta s\right)\|_1.$$

This strategy represents a fully deterministic approximation, sidestepping the biases and noise typically associated with stochastic approximation methodsLouizos et al. (2017); Srinivas et al. (2017). We employ an exponential scheduling for $\beta$, denoted by the multiplication factor $\mu = (\beta^{(E)})^{\frac{1}{E}}$ for each training epoch $e \in \{1, 2, \ldots, E\}$, to progressively increase $\beta$ from 1 to $\beta^{(E)}$. $\beta^{(E)}$ is chosen sufficiently large to ensure the binarization of the learned masks, such that $\sigma(\beta^{(E)}s^{(E)})$ closely approximates the Heaviside function $H(s^{(E)})$ by the end of training.

**Gradient normalization for seamless learning.** Nevertheless, this reformulation revealed a significant challenge: as the parameter $\beta$ increases, it induces early binarization - the premature selection of parameters to be pruned early within the training process. To counteract this saturation in the mask learning process,

we normalize the gradients of $s$ by their $l_\infty$ norm prior to each gradient update. This approach mitigates the learning plateau often observed in previous studies Savarese et al. (2019), thereby encouraging a more dynamic and seamless progression of mask parameter learning. Since the magnitude of gradient updates vary significantly across layers, this leads to varying saturation rates in mask learning as $\beta$ increases. Using the normalized gradient method and selecting distinct values of $\beta_l^{(E)}$ for each layer $l \in \{1, 2, \dots, L\}$ ensures that $s$ learns at comparable rates across the layers. This prevents premature saturation in layers that would otherwise dominate due to larger gradient updates, promoting a smoother optimization landscape.

### 4.3 On the Path to Winning Tickets: Final Stop - Controlling Sparsity

It is important to recognize that while standard regularization, specifically the term $\|\sigma\left(\beta s\theta^{(0)}\right)\|_2$, promotes sparsity, it lacks the precision necessary for fine control over the sparsity levels, particularly when the goal is to prune a predetermined fraction of parameters. On the other hand, constraint-based optimization methods, despite offering theoretical guarantees, often fall short in achieving the desired sparsity levels and are observed to compromise training accuracy. Subsequently, the objective in the final step is to incorporate effective sparsity control not only to avoid the risk of layer collapse, as described by Tanaka et al. (2020) - which refers to the premature pruning of an entire layer, rendering the network untrainable - but also to manage sparsity in a way that ensures the unpruned network post-extraction is conducive to stable learning, comparable to that of the original, dense network.

**Dynamic and progressive scoring.** We introduce a custom-designed element-wise scoring function $r(\cdot)$ inspired by Edge-Popup (EP) (Ramanujan et al., 2020) and GEM-MINER (Sreenivasan et al., 2022). This function enables efficient sparsity control offering flexibility for both global and gradual pruning across epochs. Each parameter $s$ is assigned a score reflecting its significance. We employ a strategic exponential function, $1 - e^{-\hat{c}e}$, where $\hat{c} = \frac{1}{E} \ln \frac{1}{1-S}$, to progressively prune parameters based on the assigned scores throughout training. This function determines the pruning percentile per epoch, ensuring the desired final sparsity level $S$ is reached by the final epoch $E$.

1. **Sorting the Scores**: Parameters are sorted according to $p_{\text{sorted}} \leftarrow \text{Sort}(\sigma(\beta\mathbf{s}))$.

2. **Selecting the Bottom Fraction of Scores**: The bottom fraction of scores, $p_{\text{bottom}}$, is determined by $p_{\text{bottom}} \leftarrow \text{Bottom-}(1 - e^{-\hat{c}e})$ fraction of $p_{\text{sorted}}$.

3. **Defining the Scoring Function**: The scoring function $r$ is updated as $r \leftarrow r \odot \mathbf{1}p_i \notin p\text{bottom}$.

4. **Setting the Effective Parameters**: The effective parameters are set as $\theta = \theta \odot r(\sigma\left(\beta s\right))$.

.

The scoring function retains parameters with higher absolute magnitudes. This ranking is then used to globally select $e^{-\hat{c}e}$ percentile of parameters to retain across all network layers in epoch $e$. $f_L^s\left(x; r(\sigma\left(\beta^{(e)}s^{(e)}\right)) \odot \theta^{(0)}\right)$ symbolizes the resultant subnetwork at epoch $e$. During each iteration, the forward pass involves effectively utilizing a subnetwork of the original network. Given the inherent non-differentiability of the scoring function $r(\sigma\left(\beta s\right))$ we employ Straight Through Estimator (STE) to facilitate backpropagation as if the scoring function were an identity function. This application of STE serves as an effective workaround, ensuring a seamless computational process despite the non-differentiable nature of the scoring function. It is to be noted that using this custom designed scoring function allows gradients to flow through even the pruned parameters and allows the possibility for these pruned parameters to be revived in the future iterations. Moreover, the gradual pruning process embedded within the scoring operation facilitates finer-grained decisions regarding parameter retention.

**Incremental freezing.** Alongside the scoring function, our framework integrates an incremental freezing mechanism. This mechanism sets the effective parameters as $\theta = \theta \odot r(\sigma(\beta s)) \odot q$. While the scoring function is actively applied in each training iteration, the freezing operation is distinctively implemented at regular intervals, specifically every $V$ epochs. Unlike the scoring function's dynamic approach, the freezing mechanism plays a crucial role in blocking the gradients to pruned scores, thus entirely preventing the

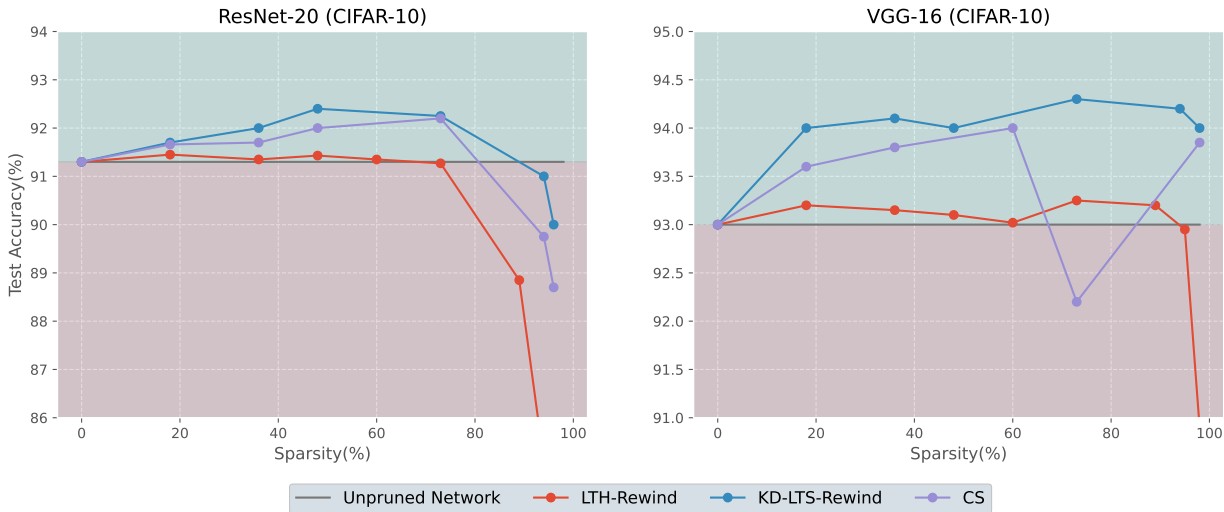

Figure 5: Comparative evaluation of KD-LTS winning ticket performance in VGG-16 and ResNet-20 architectures on the CIFAR-10 dataset, contrasted with prominent early epoch pruning methods, namely LTH-Rewind and CS.

parameters' influence on subsequent training iterations and negating the possibility of their future revival. Selectively freezing certain parameters is instrumental in allowing each subnetwork to progressively learn at every iteration, leveraging the learning dynamics of only the unpruned network. This step is crucial for crafting a subnetwork that ensures the remaining unpruned weights are capable of being trained as efficiently and seamlessly as during the initial training phase of the original, dense network.

The complete pseudocode of this carefully crafted KD-LTS framework is presented in Algorithm 1. Notably, Allen-Zhu and Li (2020) discovered that an ensemble of independently trained neural networks can provably enhance test accuracy, and this improvement can be distilled into a single student model by training it to match the ensemble's output - a feat that traditional training methods, which rely solely on true labels, could not achieve. Building upon this foundation, we extend this unique characteristic of knowledge distillation further into another novel domain, i.e., network pruning. By transferring heterogeneous information from an ensemble of teacher networks into a single student network, we identify sparse, trainable subnetworks at initialization that outperform the original, densely connected student network - a feat that existing pruning-at-initialization techniques, relying solely on true labels, have not accomplished.

## 5   Experimental Observations

We conducted our experiments using PyTorch on NVIDIA A100 GPUs, spanning a diverse set of standard image classification architectures: VGG-16, LeNet-300-100, ResNet-18, ResNet-20, ResNet-14, and a 6-layer convolutional network. The conducted trials used popular datasets: MNIST, CIFAR-10, CIFAR-100, and Tiny-ImageNet. Optimal training hyperparameters were employed to ensure standard and comparability across experiments in the literature, as illustrated in Tables 4, 5, 6 and 7. For thorough benchmarking, the approach was compared to state-of-the-art pruning methods: SNIP, GraSP, SynFlow, ProsPr, GEM-MINER, LTH, LTH-Rewind, CS, SS, Edge-Popup, Smart Ratio, and Iterative SNIP. Comprehensive details regarding the experimental setup, including data pre-processing, network architectures, implementation of comparative techniques, optimizer settings, and configurations for the knowledge distillation framework, are provided in Section G of the appendix.

**Assessment criteria and metrics.** Three core metrics form the foundation of our analysis: • **Accuracy:** Measures post-training precision of subnetworks, evaluating the pruning process's impact on performance preservation or enhancement. • **Parameter efficiency:** Indicates the pruned network's ability to maintain performance with significantly fewer parameters. Expressed as the percentage of weights pruned, this reflects

Table 2: Comparative evaluation of winning tickets extracted at initialization across diverse experiments: The table uses an ✗ symbol to indicate the absence of sparsest winning tickets, signifying a technique's failure to extract winning tickets at sparsity levels under 20%. 'T.A.' represents Test Accuracy.

| Criteria | Method | VGG-16 | | ResNet-20 | | LeNet-300-100 | | ResNet-18 | |
|---|---|---|---|---|---|---|---|---|---|
| | | T.A. (%) | Sparsity (%) | T.A. (%) | Sparsity (%) | T.A. (%) | Sparsity (%) | T.A. (%) | Sparsity (%) |
| Sparsest winning ticket | LTH | 93 | 20 | ✗ | 20 | 98 | 20 | 51 | 36 |
| | SNIP | 93 | 20 | ✗ | 20 | 98 | 20 | 51 | 73 |
| | GraSP | ✗ | ✗ | ✗ | ✗ | ✗ | ✗ | ✗ | ✗ |
| | ProsPr | 93 | 20 | 91.3 | 20 | N/A | N/A | 51 | 60 |
| | Random | 93 | 20 | ✗ | ✗ | ✗ | ✗ | ✗ | ✗ |
| | SynFlow | 93 | 20 | ✗ | ✗ | ✗ | ✗ | 51 | 73 |
| | GEM-MINER | 93 | 90 | 91.3 | 25 | N/A | N/A | 51 | 80 |
| | Iterative SNIP | 93 | 20 | ✗ | ✗ | 98 | 20 | 51 | 60 |
| | KD-LTS | 93 | 95 | 91.3 | 65 | 98 | 80 | 51 | 87 |
| Best performing winning ticket | LTH | 93 | 20 | 91.15 | 20 | 98 | 32 | 51 | 36 |
| | SNIP | 93 | 20 | 91.25 | 20 | 97.96 | 32 | 51 | 73 |
| | GraSP | 92.5 | 20 | 90.45 | 20 | 97.8 | 20 | 50 | 20 |
| | ProsPr | 93.1 | 48 | 91.45 | 20 | N/A | N/A | 51 | 60 |
| | Random | 93 | 20 | 91.15 | 20 | 97.92 | 20 | 50.1 | 20 |
| | SynFlow | 92.9 | 20 | 91.05 | 20 | 97.92 | 32 | 51.2 | 20 |
| | GEM-MINER | 93 | 55 | 91.3 | 20 | N/A | N/A | 51.2 | 55 |
| | Iterative SNIP | 93 | 20 | 91.15 | 20 | 98 | 20 | 51.2 | 73 |
| | KD-LTS | 93.7 | 40 | 91.8 | 40 | 98.33 | 45 | 51.4 | 45 |

the network's compactness. • **Search cost:** Quantifies the computational effort to find pruned subnetworks, measured in epochs needed to reach a desired sparsity. This reflects the method's practical efficiency.

This section compares the performance of KD-LTS winning tickets in sparse, large-scale settings, both at initialization and early training stages. In this assessment, special attention is given to two distinct types of subnetworks that demonstrate the efficacy of KD-LTS as an efficient pruning method: • **Sparsest winning ticket:** Refers to the most aggressively pruned subnetwork that, when trained independently from an early iteration, matches or exceeds the performance of the fully-trained dense network. • **Best performing winning ticket:** Denotes the subnetwork, that when trained in isolation, achieves the highest performance across all pruned networks, regardless of sparsity. These comparisons showcases KD-LTS's ability to maximize parameter efficiency, maintain accuracy, and minimize search costs, showcasing its real-world potential. Table 2 presents results for both sparsest and best-performing winning tickets across various pruning methods.

## 5.1 Winning Tickets That Really Win the Jackpot and Beyond

**Winning tickets at initialization.** To fully understand the potential of KD-LTS and initialization pruning, a comprehensive comparitive analysis must extend beyond just test performance against existing initialization methods. This analysis should evaluate search efficiency, parameter reduction, and accelerated learning against both dense networks and early-phase pruning methods. Such a comparison, we believe, would facilitate a research shift from early-stage to initialization pruning.

Figures 3, 22, and 6 provide crucial insights into these capabilities for KD-LTS winning tickets. Figures 3 and 22 illustrate the sparsity-accuracy trade-off of KD-LTS's initialization-derived winning tickets. They benchmark KD-LTS against established early pruning and pruning-at-initialization techniques across various architectures and datasets (VGG-16/CIFAR-10, LeNet-300-100/MNIST, ResNet-18/Tiny-ImageNet, ResNet-20/CIFAR-10, and ResNet-20/CIFAR-100). Figure 6 demonstrates the superior learning efficiency of KD-LTS winning tickets across training epochs at various sparsity levels, benchmarked against dense networks.

Table 3: Comparative evaluation of winning tickets extracted at the third epoch across diverse experiments.

| Criteria | Method | VGG-16 | | ResNet-20 | |
|---|---|---|---|---|---|
| | | Test Accuracy (%) | Sparsity (%) | Test Accuracy (%) | Sparsity (%) |
| Sparsest winning ticket | CS | 93 | 97 | 91.3 | 80 |
| | LTH-Rewind | 93 | 89 | 91.3 | 73 |
| | KD-LTS | 93 | 98 | 91.3 | 90 |
| Best performing winning ticket | CS | 94 | 60 | 92.2 | 73 |
| | LTH-Rewind | 93.2 | 73 | 91.45 | 20 |
| | KD-LTS | 94.2 | 94 | 92.4 | 48 |

The following summary highlights key findings on KD-LTS winning tickets at initialization, emphasizing advantages in test performance, parameter efficiency, and learning efficiency over previous benchmarks:

---

**Key Findings**

- **Parameter efficiency**: KD-LTS significantly outperforms previous state-of-the-art results in sparsity across various models and datasets: VGG-16/CIFAR-10: 95% sparsity achieved (vs. previous 90%), ResNet-20/CIFAR-10: 65% sparsity (vs. previous 25%), LeNet-300-100/MNIST: 80% sparsity (vs. previous 20%) and ResNet-18/Tiny-ImageNet: 87% sparsity (vs. previous 80%). • **Test efficiency**: Our method demonstrates superior test accuracy compared to existing techniques: VGG-16/CIFAR-10: 94% (vs. previous 93.1%), ResNet-20/CIFAR-10: 91.8% (vs. previous 91.4%), LeNet-300-100/MNIST: 98.33% (vs. previous 98%) and ResNet-18/Tiny-ImageNet: 51.4% (vs. previous 51.2%). • **Learning efficiency**: Subnetworks identified by KD-LTS exhibit significantly accelerated learning, converging on an average 1.2 times faster than the dense network, indicating a substantial improvement in training speed and efficiency.

---

**Supplementary benefit: Enhancing performance through early-phase extraction.** Our method represents a multifaceted approach to identifying sparse networks. Emphasizing the aspect of performance, particularly in terms of accuracy, we showcase a complementary advantage: the winning tickets, identified not only at initialization but also during early training epochs using KD-LTS, exhibit state-of-the-art performance. While extraction of winning tickets at initialization follows the protocol outlined in Algorithm 1, a slight adjustment is employed for extracting winning tickets at the early training phase: In this variant, the student model's weights and biases are trained up to the third epoch, mirroring the training settings used for subnetworks post-extraction.

The results presented in Table 3 and Figure 22 aim to showcase the trade-off between resource conservation (in terms of storage and computational costs saved by extracting winning tickets at initialization) and the achievement of performance gains (in terms of accuracy and sparsity achieved by extracting winning tickets at the third epoch) across experiments. This equilibrium demonstrates the versatility of our method, highlighting its utility in various applications with diverse requirements and specifications, across the trade-off continuum mentioned. For further insights into early-training extraction, Figures 5 and 2 provide detailed comparisons of KD-LTS's performance in ResNet-20/CIFAR-10 and VGG-16/CIFAR-10 settings. Figure 5 focuses on the sparsity/accuracy efficiency, while Figure 2 emphasizes search cost efficiency.

---

**Key Findings**

- **Parameter efficiency**: KD-LTS significantly outperforms LTH-Rewind in terms of sparsest winning tickets extracted at early epochs: VGG-16/CIFAR-10: 98% sparsity (vs. LTH-Rewind 89%) and ResNet-20/CIFAR-10: 90% sparsity (vs. LTH-Rewind 73%). • **Test efficiency**: The best-performing subnetworks achieve superior test accuracy compared to LTH-Rewind: VGG-16/CIFAR-10: 94.2% (vs. LTH-Rewind 93.2%) and ResNet-20/CIFAR-10: 92.4% (vs. LTH-Rewind 91.45%). • **Search efficiency**: This approach is significantly faster in identifying high-performance, sparse winning tickets: 3-4x faster than Continuous Sparsification (CS), 19x faster than LTH-Rewind. • **Initialization vs early epoch pruning**: Winning tickets extracted from early epoch using KD-LTS demonstrate improved performance over initialization-derived counterparts: Test accuracy increase of 0.6-0.9% and sparsity reduction of 8-35%. • **Comprehensive analysis of training and optimization landscape**: Subnetworks identified by our technique at initialization display learning patterns akin to those identified at early epochs by current methods. This similarity substantiates their superior performance and positions KD-LTS as a viable alternative to early epoch pruning.

---

**Assessment of the training landscape.** We present an in-depth analysis on the intricate dynamics of KD-LTS' subnetworks extracted at initialization against those extracted in early epochs, and with other pruning at initialization methods, highlighting the superior performance of KD-LTS in fostering stable training and enhanced generalization. In Figure 7, we illustrate the evolution of ResNet-20 during the initial training phase, with a specific focus on the first 2112 iterations (equivalent to 6 epochs). Our first line of examination

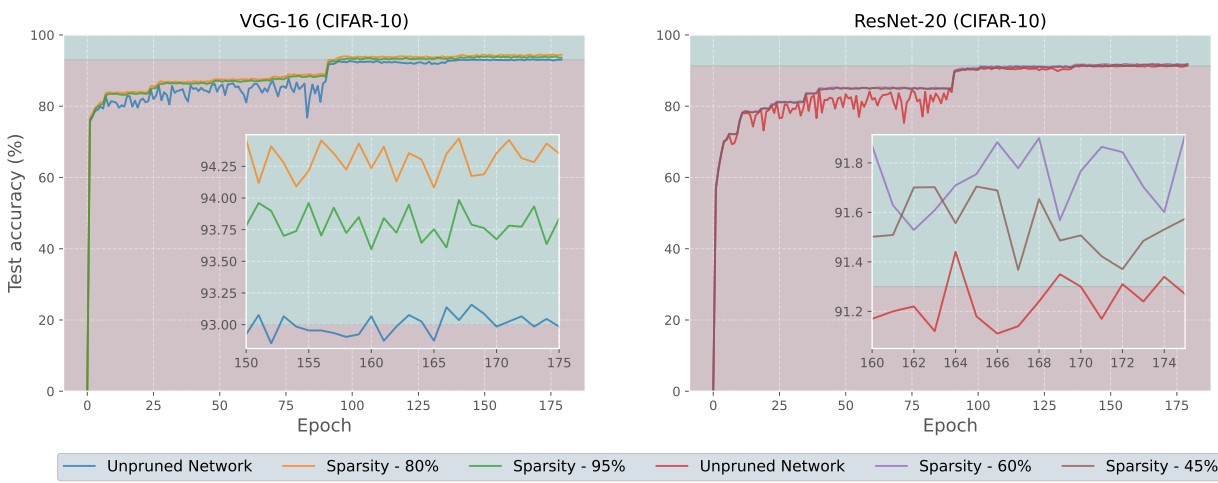

Figure 6: Learning efficiency of KD-LTS' winning tickets juxtaposed with the dense network.

includes a range of pivotal telemetrics, encompassing average weight magnitude, the proportion of weights that change sign after initialization, gradient magnitude, $L_2$ distance measuring the deviation of weights from their initial to current values, and the cosine similarity between initial and current weights. The research by Frankle et al. (2020a) points out that large learning rates or complex experimental settings can disrupt the optimization process by excessively perturbing weights, resulting in reduced stability to SGD noise. This phenomenon is identified in the literature as a principal reason why existing initialization pruning techniques struggle to discover winning tickets at the onset of training and in contrast why early-phase pruning methods tend to be successful in this regard. The comprehensive telemetric assessment highlight the analogous training behavior of KD-LTS' subnetworks extracted at initialization to those identified in early epochs by current methods, thereby underlining their training stability and transformative potential.

**Examination across the optimization landscape.** Second line of exploration plotted in Figure 8 includes the concept of gradient predictiveness, as elucidated in Santurkar et al. (2018), this metric refers to the $L_2$ change in gradient when moved along the gradient direction for different learning rates. In the context of KD-LTS , the findings showcase a smoother gradient predictiveness akin to the characteristics observed in subnetworks identified through early pruning techniques. The smoother gradient predictiveness suggests a more predictable and stable gradient evolution in the training trajectory. Utilizing PyHessian, we plot the Hessian eigen spectral density, examining the Hessian properties of subnetworks extracted using KD-LTS in comparison to standard methods. The results demonstrate a notable capacity to evade saddle points, similar to networks identified by early pruning methods. By measuring the variation in loss at each training step, illustrated as a shaded region in our graphs, we observe how the loss fluctuates across iterations. This metric not only indicates enhanced Lipschitz interpretability similar to early pruning methods. Following the insights from (Jastrzębski et al., 2017; Sankar et al., 2021), a reduced value of Hessian trace correlates with a broader optimum width, indicative of flatter minima and consequently, enhanced generalization capabilities. In an effort to substantiate the superior generalization of KD-LTS subnetworks, we employ the 3D loss landscape visualization technique, using tools from (Li et al., 2018) and the Paraview application. This visualization at the optimum, presented in Figure 9, provides a compelling representation of our subnetworks' generalization strengths. Complementing this, Figures 27, 28 and 29 extends our analysis to additional networks.

Through these findings, our modest hope is to reinvigorate the concept of pruning at initialization and shift the research focus from extracting winning tickets from early epochs towards initialization, thereby opening up prospects for a significant decrease in both computational and storage complexities right from the start of training.

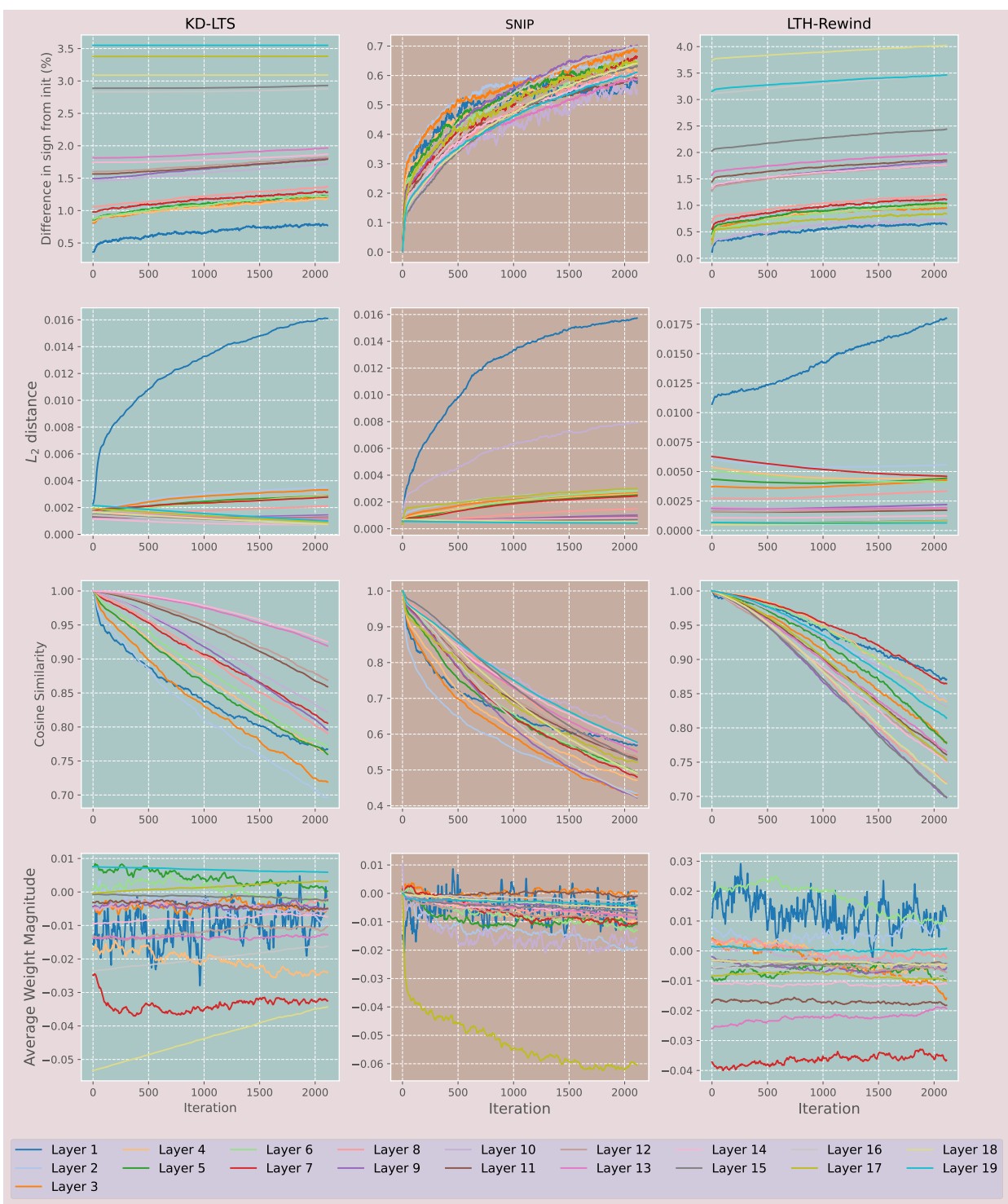

Figure 7: Training dynamics of weight parameters in ResNet-20 subnetworks on CIFAR-10 across the first 2112 iterations, for subnetworks extracted using KD-LTS, SNIP, and LTH-Rewind.

## 6   Validating the Quality of Winning Tickets

This study undertakes a comprehensive re-evaluation of LTH and addresses limitations of existing pruning-at-initialization techniques, introducing the KD-LTS framework. The findings of Frankle and Carbin (2018)

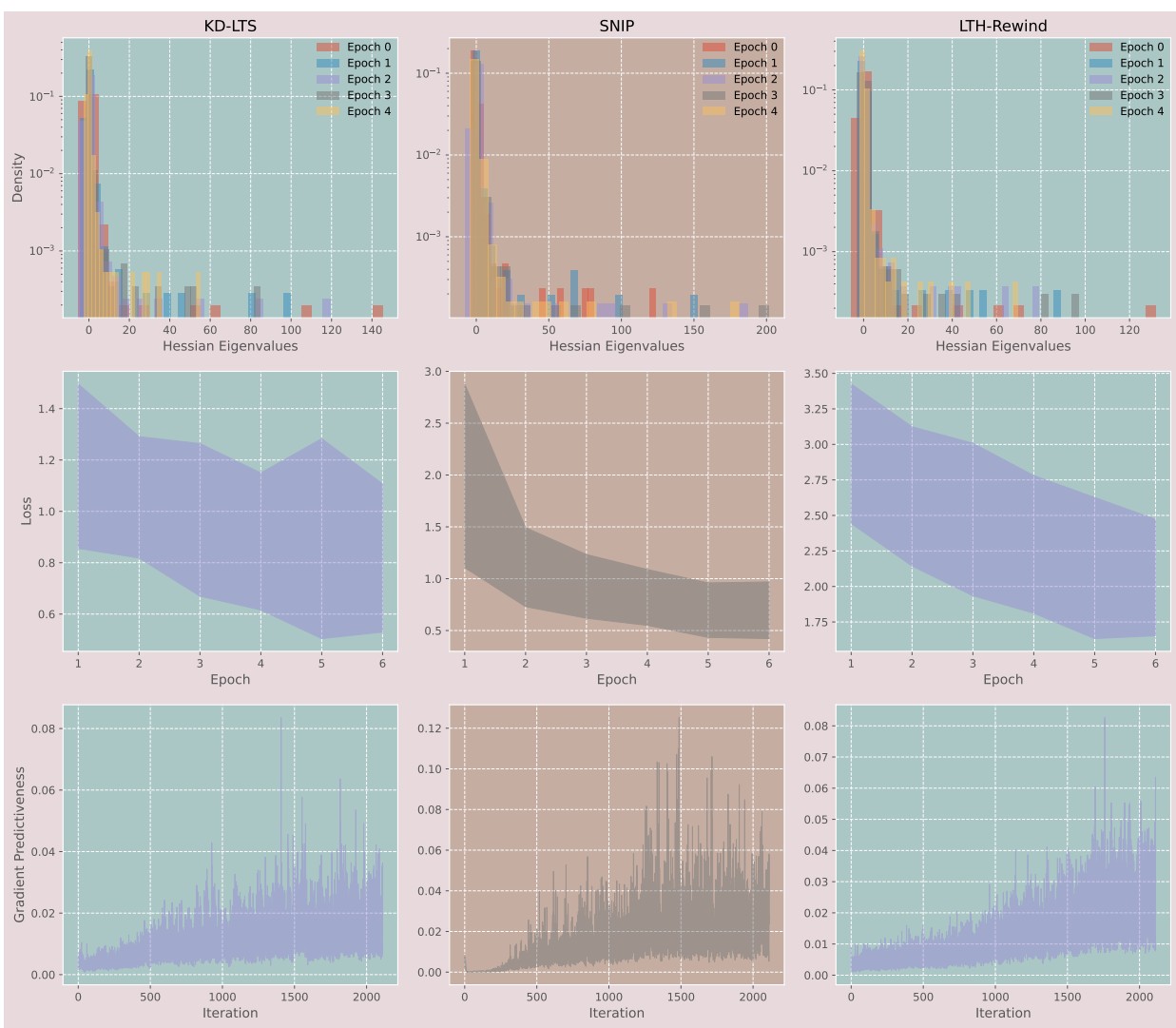

Figure 8: Observation of the optimization landscape through the lens of key Metrics in ResNet-20 subentworks trained on CIFAR-10 across the first six epochs.

revealed that the subnetworks extracted using prevailing pruning-at-initialization techniques are often susceptible to SGD noise and various random perturbations. This susceptibility was characterized in terms of the stability (Frankle et al., 2020a) and sensitivity (Frankle et al., 2020b) of the subnetworks. It should be noted that these assessments were critical in ascertaining why prevalent pruning-at-initialization techniques, including IMP, fall short in effectively identifying winning tickets and have led to a shift in the focus of pruning towards the early epochs. It's essential that research efforts refocusing the pruning techniques toward the initialization stage should involve subjecting the extracted subnetworks to established tests and quality assessments. Consequently, our investigation extends beyond merely showcasing the superior performance of KD-LTS identified winning tickets; it also encompasses a rigorous validation of their credibility. We begin by providing formal definitions of stability and sensitivity analyses, followed by articulating a strict definition of LTH to promote standardization in works on LTH. Our results demonstrate that KD-LTS subnetworks align with this exacting definition of LTH, which we hope offers a promising path to revitalize pruning at initialization.

**Stability analysis.** Stability analysis in a subnetwork, specifically $f_L(x; m^{(E)} \odot \theta^{(0)})$ with initial weights $\theta^{(0)}$, involves assessing its response to SGD noise. This is achieved by creating two copies of the network and training each with different samples of SGD noise, such as through varying data order and data augmentation

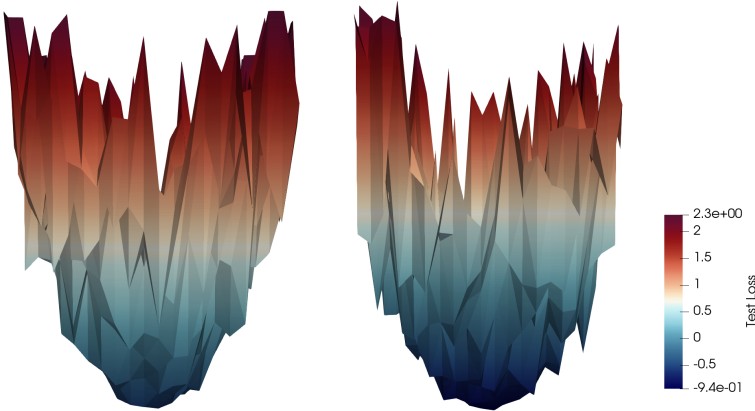

Figure 9: Visualization of the loss landscape at the optimum for ResNet-20 subnetworks, trained on CIFAR-10, with subnetworks extracted via KD-LTS (left) and LTH-Rewind (right).

techniques. The resulting networks, $f_L(x; m^{(E)} \odot \theta^{(E)'})$ and $f_L(x; m^{(E)} \odot \theta^{(E)''})$, are then analyzed for stability. The test error of a network is denoted by $\varepsilon(f_L(x; m \odot \theta))$. For a given point $\zeta$ within the range $[0, 1]$, the error $\varepsilon_\zeta(f_L(x; m^{(E)} \odot \theta^{(E)'}), f_L(x; m^{(E)} \odot \theta^{(E)''}))$ is computed by interpolating between $\theta^{(E)'}$ and $\theta^{(E)''}$, thus forming a network $f_L(x; m^{(E)} \odot (\zeta\theta^{(E)'} + (1 - \zeta)\theta^{(E)''}))$ along the linear path connecting the resulting networks $f_L(x; m^{(E)} \odot \theta^{(E)'})$ and $f_L(x; m^{(E)} \odot \theta^{(E)''})$.

The maximum error observed during this interpolation, denoted as $\varepsilon_{\mathrm{sup}}$, and the mean error, denoted as $\bar{\varepsilon}$, are key metrics to assessing stability. With this background, we define stability in the following terms:

**Definition 6.1** (Stability). A subnetwork, denoted as $f_L(x; m^{(E)} \odot \theta^{(0)})$, is deemed 'stable' with respect to SGD noise if it exhibits linear mode connectivity between the trained instances, $f_L(x; m^{(E)} \odot \theta^{(E)'})$ and $f_L(x; m^{(E)} \odot \theta^{(E)''})$. Stability is characterized by the difference between the maximum error measure, $\varepsilon_{\mathrm{sup}}(f_L(x; m^{(E)} \odot \theta^{(E)'}), f_L(x; m^{(E)} \odot \theta^{(E)''}))$, and the mean error measure, $\bar{\varepsilon}(f_L(x; m^{(E)} \odot \theta^{(E)'}), f_L(x; m^{(E)} \odot \theta^{(E)''}))$, observed along the linear path connecting the trained instances and is given by

$$\varepsilon_{\mathrm{sup}}(f_L(x; m^{(E)} \odot \theta^{(E)'}), f_L(x; m^{(E)} \odot \theta^{(E)''})) - \bar{\varepsilon}(f_L(x; m^{(E)} \odot \theta^{(E)'}), f_L(x; m^{(E)} \odot \theta^{(E)''})) \leq \Delta,$$

with $\Delta$ representing an acceptable threshold of deviation. A network is classified as 'stable' when this difference is within the threshold, indicating minimal deviation due to SGD noise. Conversely, a substantial deviation classifies the subnetwork as 'unstable', demonstrating its susceptibility to SGD noise. An allowance for a tolerance margin is included to account for noise and error variations in the analysis.

**Sensitivity analysis.** Sensitivity analysis encompasses a range of perturbation studies, each designed to assess the response of the subnetworks identified by various early pruning and initialization pruning methods under various altered conditions. Given a subnetwork $f_L(x; m^{(E)} \odot \theta^{(0)})$ with initial weight parameters $\theta^{(0)}$ and trained masks $m^{(E)}$, the perturbation procedures include:

- **Random reinitialization**: Random reinitialization refers to the procedure in which $\theta^{(0)}$, are reset randomly, resulting in a new set of initial weights $\theta^{(0)'}$. These reinitialized weights are then combined with $m^{(E)}$, to form a randomly reinitialized subnetwork, expressed as $f_L(x; m^{(E)} \odot \theta^{(0)'})$. The test accuracy achieved by training this randomly reinitialized network is denoted as $A_{\mathrm{RR}}$.

- **Inversion**: Inversion represents the operation of sign inversion of these weights, followed by the application of $m^{(E)}$, yields a perturbed subnetwork. This subnetwork, featuring sign-inverted weights, is represented as $f_L(x; m^{(E)} \odot -\theta^{(0)})$. The test accuracy of this subnetwork, post-training, is denoted as $A_{\mathrm{I}}$.

- **Random shuffling**: Random Shuffling is a procedure that entails the random permutation of $m^{(E)}$ to produce a shuffled variant, denoted as $m^{(E)'}$. These shuffled masks are then applied to $\theta^{(0)}$ to form a

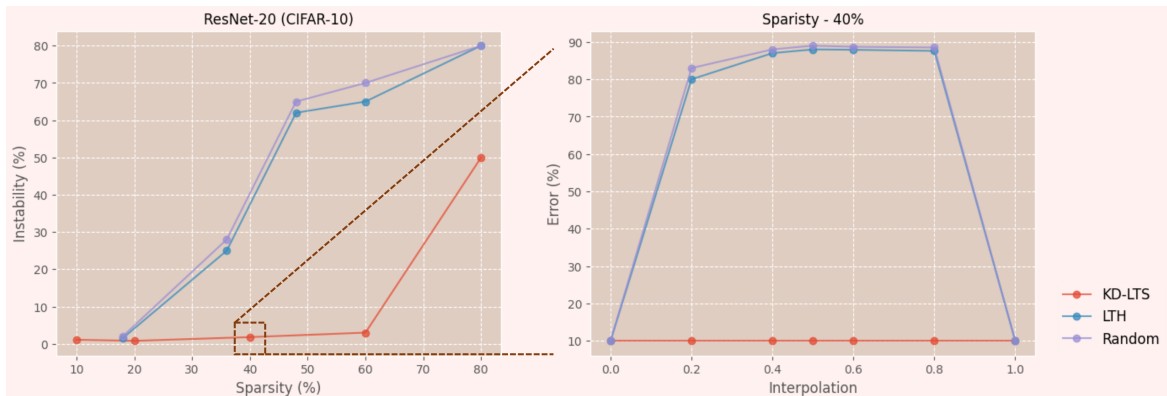

Figure 10: Assessing the indication of stability in LTH, random pruning, and KD-LTS at various sparsity levels in a ResNet-20 experiment on CIFAR-10, and evaluating linear mode connectivity, particularly at non-trivial sparsity levels.

subnetwork, expressed as $f_L(x; m^{(E)'} \odot \theta^{(0)})$. The test accuracy achieved by training this subnetwork with the randomly shuffled masks is designated as $A_{\mathrm{RS}}$.

- **Small-dense training**: Small-Dense Training refers to the process of training a compact and densely connected neural network, denoted as $f_L(x; \theta_{\mathrm{SD}})$. This network is characterized by reduced width, while maintaining the same depth as $f_L(x; m^{(E)} \odot \theta^{(0)})$, and is initialized with weight parameters $\theta_{\mathrm{SD}}$. The test accuracy achieved following the training of this compact network is represented as $A_{\mathrm{SD}}$.

**Definition 6.2** (Sensitivity)**.** A subnetwork, denoted as $f_L(x; m^{(E)} \odot \theta^{(0)})$, is considered 'sensitive' with respect to the procedures of Random Reinitialization, Inversion, Random Shuffling, and Small-Dense Training if the following condition is satisfied:

$$\min(A_{\mathrm{RR}}, A_{\mathrm{I}}, A_{\mathrm{RS}}, A_{\mathrm{SD}}) < A_S$$

signifying that the subnetwork's learning capability is significantly affected by the perturbations. Here, $A_S$ denotes the test accuracy of the trained unaltered subnetwork.

In addition to stability and sensitivity analysis, effective network training requires consideration of factors, such as appropriate learning rates and the duration of training epochs. For instance, selecting an optimal learning rate (e.g., 0.1 versus 0.01) combined with a sufficient number of training epochs (e.g., 160 versus 85) can markedly enhance test accuracy (e.g., 91.3% versus 89.0% in a ResNet-20/CIFAR-10 experiment). The literature (Frankle and Carbin, 2018; Liu et al., 2018) suggests that winning tickets often emerge under conditions of smaller learning rates and fewer epochs. As noted, inadequate training durations and suboptimal learning rates can lead to underperformance, thereby significantly undermining the potential for discovering any sparse network in the context of insufficient training. Having underscored these essential factors for assuring the quality of winning tickets, we now proceed to present the revised definition of the LTH.

**Definition 6.3** (Lottery Ticket Hypothesis)**.** Consider a subnetwork $f_L(x; m^{(E)} \odot \theta^{(0)})$, with a non-trivial sparsity $S$, where $m^{(E)}$ is a binary mask obtained through a specific pruning algorithm applied to the initial parameters $\theta^{(0)}$. After training for $E'$-epochs, let $A_{\mathrm{LT}}$ denote the test accuracy achieved by $f_L(x; m^{(E)} \odot \theta^{(0)})$. Furthermore, let $A_{\mathrm{D}}$ represent the accuracy of the densely connected network $f_L(x; \theta^{(0)})$, trained with an optimal learning rate schedule over a sufficient duration of $E$ epochs. The Lottery Ticket Hypothesis is then stated as follows: $\exists$ a subnetwork $f_L(x; m \odot \theta^{(0)})$, which when trained for $E' \leq E$ epochs, can achieve an accuracy $A_{\mathrm{LT}}$ such that $A_{\mathrm{LT}} \approx A_{\mathrm{D}}$, and $A_{\mathrm{LT}} > \max\{A_{\mathrm{RR}}, A_{\mathrm{I}}, A_{\mathrm{SD}}, A_{\mathrm{RS}}\}$, where ">" indicates a clear accuracy gap. Additionally, $\varepsilon_{\sup}(f_L(x; m^{(E)} \odot \theta^{(E)'}), f_L(x; m^{(E)} \odot \theta^{(E)''})) - \bar{\varepsilon}(f_L(x; m^{(E)} \odot \theta^{(E)'}), f_L(x; m^{(E)} \odot \theta^{(E)''})) \leq 2$. Such a subnetwork $f_L(x; m^{(E)} \odot \theta^{(0)})$, if identified, is referred to as a *winning ticket*.

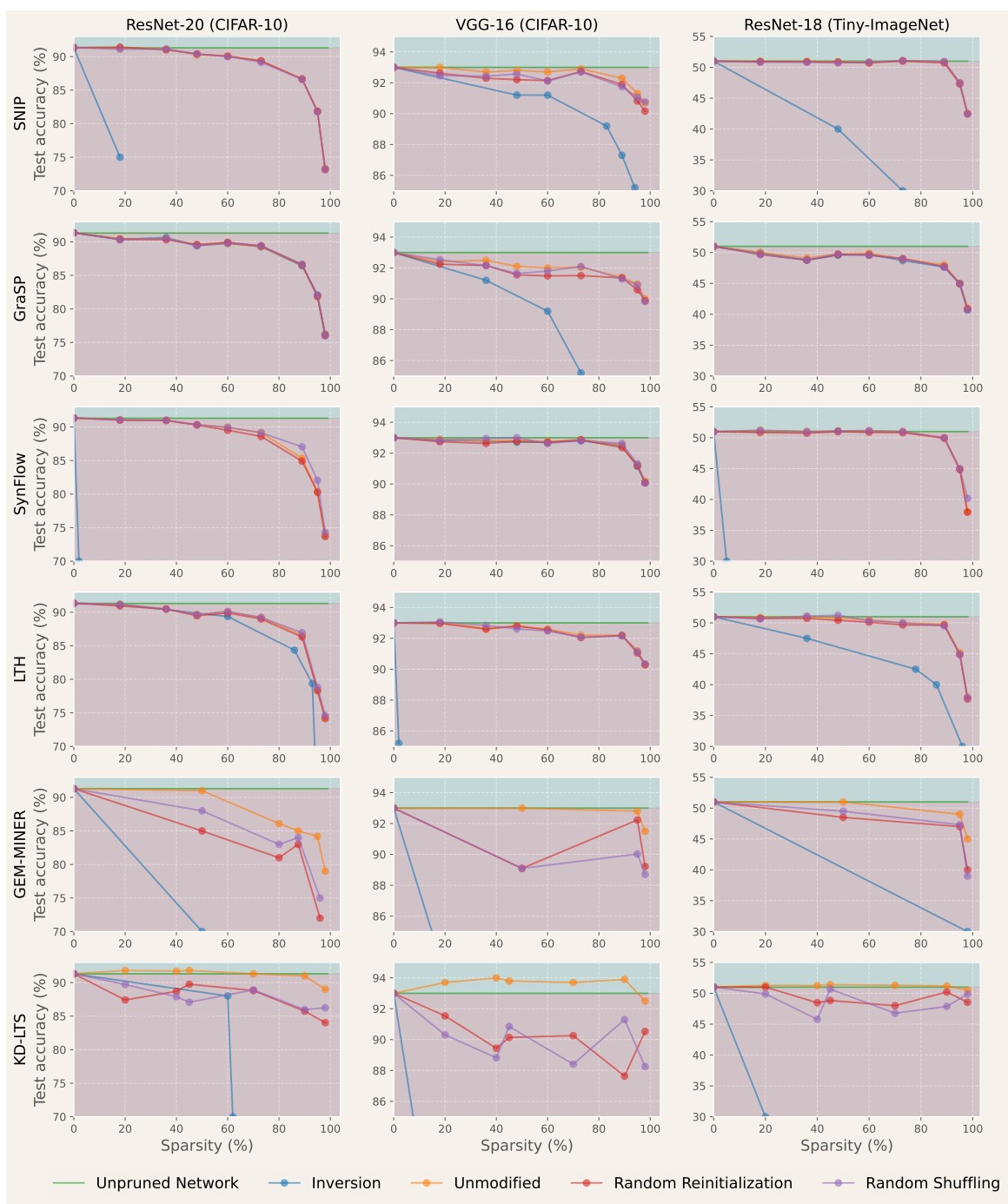

Figure 11: Perturbation studies on subnetworks extracted using LTH, SNIP, GraSP, SynFlow, GEM-MINER and KD-LTS at initialization across experiments.

Figure 10 illustrates the linear mode connectivity of KD-LTS winning tickets, verified across 30 interpolation points in a ResNet-20/CIFAR-10 setting. This analysis contrasts KD-LTS with subnetworks identified at initialization by LTH and random pruning methods. The confirmation of a linear path between trained instances of the KD-LTS identified subnetworks is consistent across experiments. Furthermore, Figure 11 elucidates the sensitivity of these winning tickets to perturbations, such as random shuffling, inversion,

and reinitialization, in contrast to subnetworks extracted by LTH, SNIP, GraSP, SynFlow, ProsPr, and GEM-MINER, spanning diverse networks and datasets.

> **Key Findings**
>
> - **Precise connection identification.** The insensitivity of existing initialization pruning methods to ablation studies, such as random shuffling, raises questions about the necessity of the fine-grained operations performed by these algorithms. This observation suggests that a more coarse-grained, per-layer approach to pruning might yield similar, if not superior, results. However, KD-LTS distinguishes itself with its precision by adeptly discerning not only the optimal percentage of weights to prune per layer but also by crucially identifying specific connections that are essential for forming a winning ticket.
> - **Randomization ambiguity exclusion.** The response of KD-LTS subnetworks to perturbations, such as inversion and reinitialization, convincingly demonstrates that randomization cannot substitute for any critical stages in the KD-LTS process. • **Assurance of search cost returns.** By adhering to an optimal training recipe and demonstrating the superior performance of KD-LTS winning tickets compared to small-dense networks, this methodology guarantees a positive return on the computational investment made in the pruning process. This establishes KD-LTS and knowledge distillation as guiding principles for a highly efficient, real-world sparse network solution, especially in resource-constrained environments.

## 7  Conclusion

The study challenges the conventional wisdom that highly efficient sparse subnetworks cannot be identified at initialization. Through the proposed KD-LTS technique, we demonstrate the feasibility of initiating pruning at the onset, particularly within demanding scenarios (deeper networks, larger datasets, and more rewarding training settings). By showcasing superior overall performance and rigorous verification of the quality of KD-LTS winning tickets, we aim to catalyze this paradigm shift, which holds immense potential to significantly reduce training costs in terms of computation, storage, energy consumption, environmental impact, and financial expenditures. Additionally, we spotlight the potential of KD-LTS in embedding specialized traits and its capacity as a practical tool for pruning, leveraging the rich history of research in knowledge distillation.

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

# Appendix

## A   Advancing Beyond Inheritance: Embracing Knowledge Distillation for Comprehensive Assessment

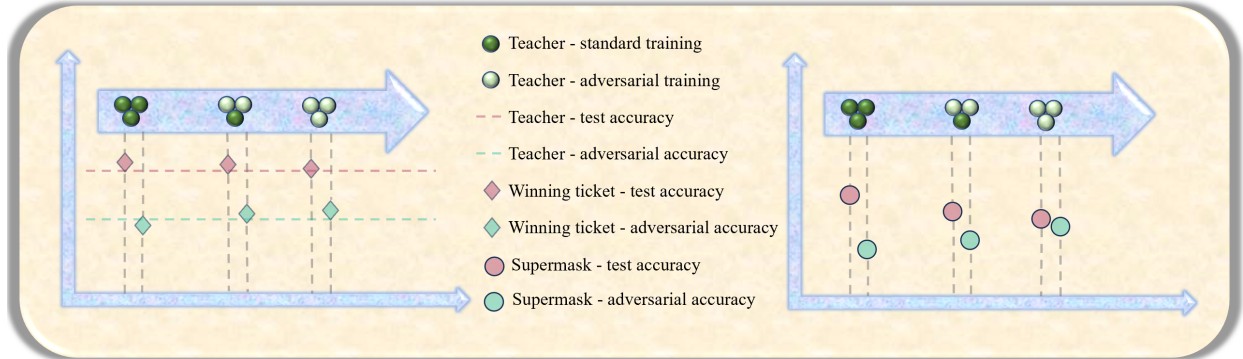

Figure 12: A sketchmap of two counter-intuitive insights: It demonstrates how the KD-LTS model can facilitate the identification of adversarially robust supermasks (on the left) and adversarially robust winning tickets (on the right), utilizing an adversarially robust teacher, all while preserving test performance.

KD-LTS framework capitalizes on a synergy between ground truth label training and the rich, heterogeneous information sourced from an ensemble of teacher networks. As highlighted in the previous section, this innovative approach extends beyond merely enhancing the test accuracy of subnetworks derived from student models. We strategically broaden the already laid out scope and utilize the knowledge distillation platform to further enhance the traits of the subnetworks. This is achieved by utilizing a carefully chosen set of teacher networks with unique attributes, as opposed to standard teacher networks.

> **Key Insight**
>
> KD-LTS uncovers an unexplored and non-intuitive connection: Knowledge distillation, when executed with teacher networks that are robust to adversarial attacks, goes beyond simply identifying robust supermasks, even when distillation is performed with standard data. These supermasks evolve into winning tickets after training, with enhanced adversarial robustness while maintaining test performance, achieved through standard training without any adversarial data. This contrasts with supermasks and winning tickets derived from standard teacher networks, underscoring the distinctive advantage of our approach.

Looking ahead, there is an intention to further explore the potential of KD-LTS framework. The aim is to endow winning tickets with greater interpretability, reinforce confidence in their predictions, bolster their overall reliability and improve transferability beyond adversarial robustness. This will be achieved by using teacher networks that exhibit qualities such as interpretability, confidence, reliability, and those trained on source optimizer, dataset, model or application, to enhance transferability.

> **Key Findings**
>
> • **Transcending merely savoring overall performance: Actively embedding traits in winning tickets.** KD-LTS subnetworks derived using standard teacher models exhibit a 2-3% enhancement over dense networks and subnetworks extracted using early-phase pruning. Figure 13 demonstrates that winning tickets derived from student networks, which incorporate lottery-ticket information from adversarially robust teacher networks, achieve an additional 3% improvement in adversarial robustness. This increment is in addition to the 2-3% improvement already observed with standard teacher models. Furthermore, we have observed a 2-4% increase in the robustness of supermasks.

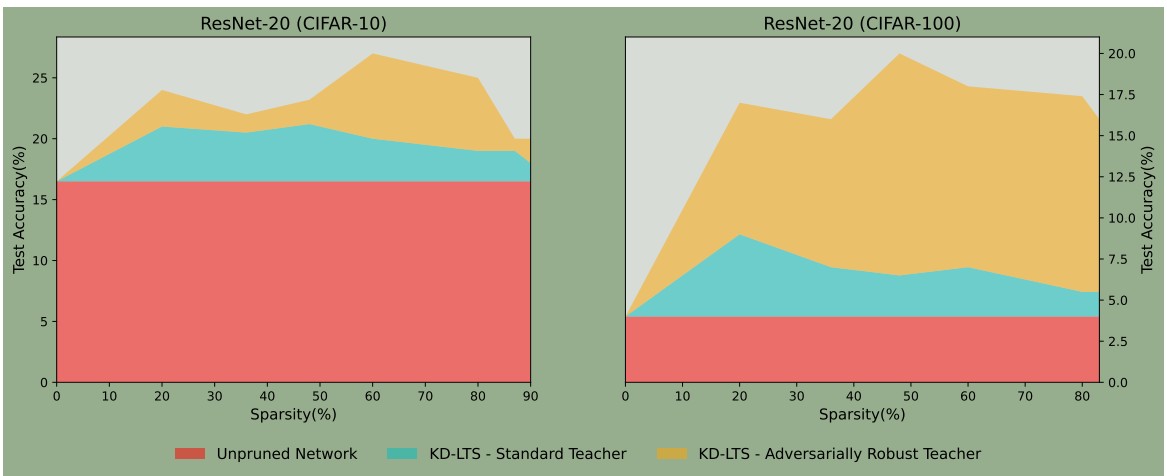

Figure 13: Adversarial accuracy of winning tickets obtained in a ResNet-20 network across datasets CIFAR-10 (left) & CIFAR-100 (right) using standard teacher and adversarially robust teacher.

## B Ablation Studies: Key Observations and Insights for Future Research Directions

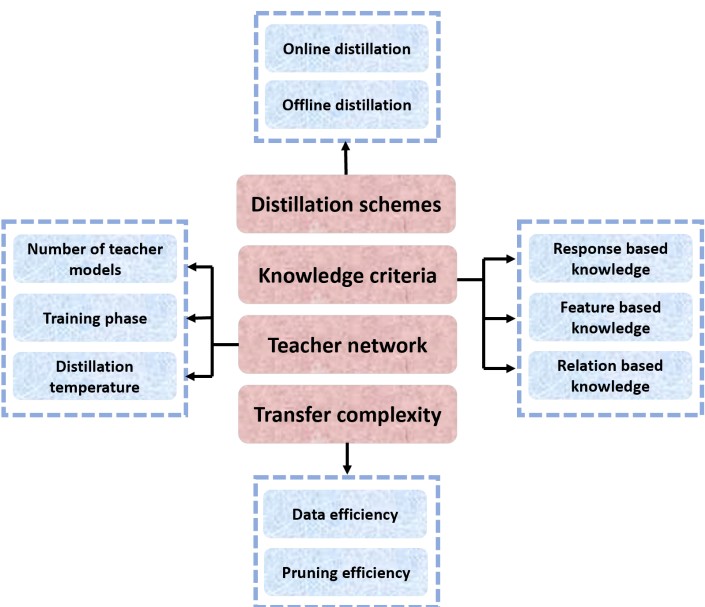

Figure 14: Dissection on the major components of KD-LTS framework.

This section details ablation studies within the KD-LTS framework, highlighting variations across knowledge categories, distillation schemes, teacher model configurations, and the complexities in knowledge transfer. These analyses are conducted using a modest experimental setup, leveraging the LeNet-300-100 architecture trained on the MNIST dataset. Although extending these studies beyond this scale is outside this paper's purview, the ablation studies presented aim to map out multifaceted avenues for future research within the KD-LTS platform. The exploration not only seeks to provide in-depth insights into the framework's effectiveness as a research tool - demonstrating the feasibility of model pruning at initiation - but also to affirm its practical value in identifying effective winning tickets from the outset.

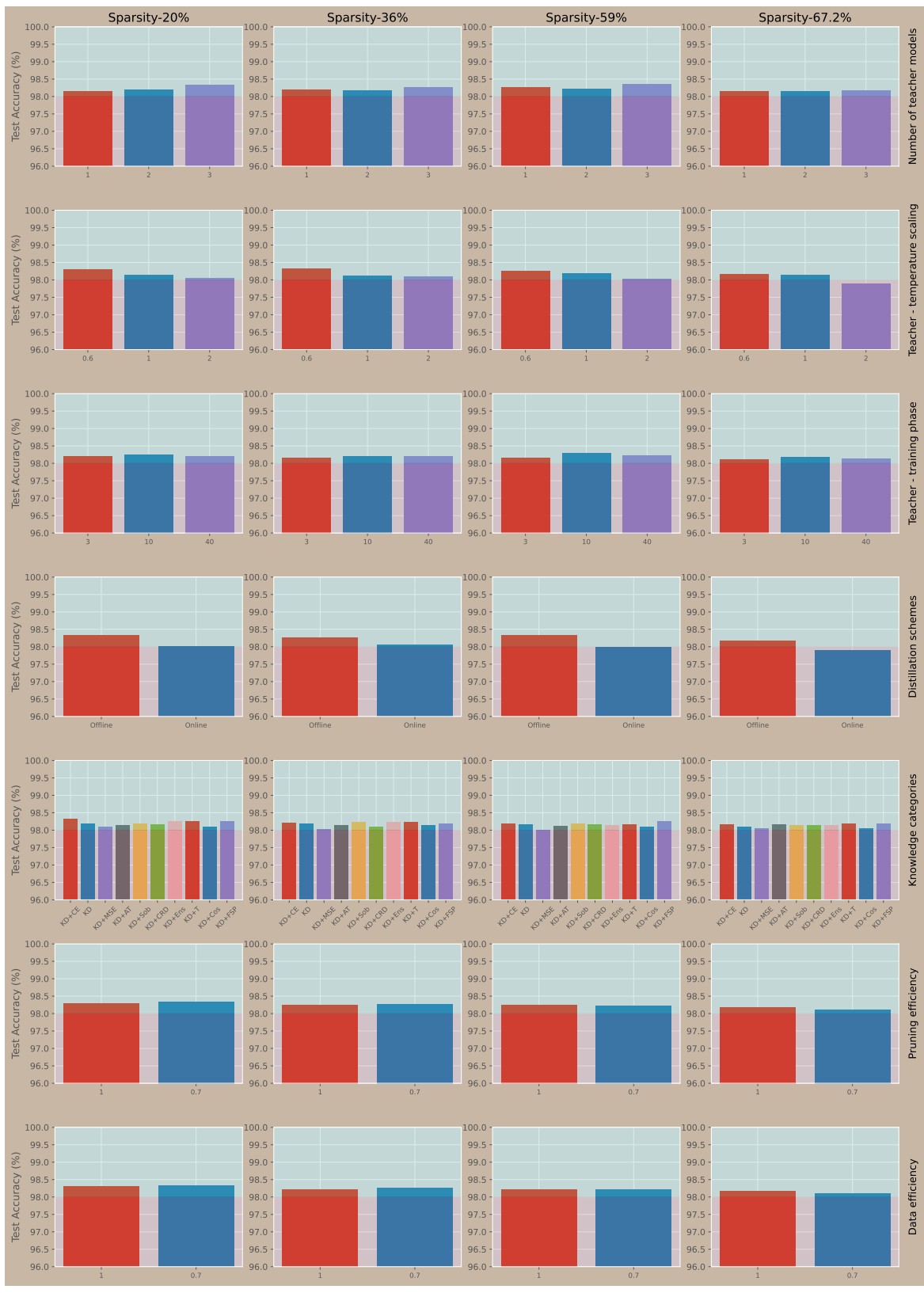

Figure 15: Ablation studies on key componenets of KD-LTS framework in a LeNet-300-100 (MNIST) environment.

- **Number of teacher models.** Figure 15 demonstrate that leveraging representational and structural insights from multiple teacher models helps in identifying high-performing winning tickets, notably in the high sparsity regime. Interestingly, the accuracy improvements achieved through a multi-teacher distillation strategy in the rare sparsity regime, compared to employing a single teacher model, are marginal. This finding suggests that a singular teacher model is capable of encoding the crucial lottery-ticket information necessary for identifying winning tickets. To facilitate the transfer of knowledge from an ensemble of teacher models, we initially employed a straightforward averaging of their responses and features, a technique that incurs a computational cost of $\mathcal{O}(n)$. Conversely, by implementing a strategy that involves randomly selecting a single teacher model at each iteration, we significantly reduced the computational requirements to $\mathcal{O}(1)$, while still tapping into the diversity offered by the ensemble. This computationally efficient approach yielded results comparable to those of the vanilla multi-teacher averaging in terms of winning ticket performance, underscoring the efficacy of simpler distillation methodologies in the context of KD-LTS.

- **Training phase of teacher models.** Substantial research (Frankle et al., 2020c; Gur-Ari et al., 2018; Achille et al., 2017) has dissected the complexities of DNNs across various training stages. This study extends the exploration to assess the impact of different training stages on the identification of high-performance winning tickets. By examining the teacher model at distinct phases of training, this investigation aims to pinpoint the optimal stage for transferring critical information to the student network, thereby facilitating the efficient discovery of winning tickets. Remarkably, it was found that teacher models at intermediate training stages—specifically around the 10th epoch—yielded winning tickets with superior performance compared to those at their early 3rd epoch or their later 40th epoch. This observation aligns with findings from traditional knowledge distillation research on model checkpoints, as reported by Wang et al. (2022). Surprisingly, this observation suggests that even teacher networks in the early stages of training possess enough valuable information for the effective extraction of winning tickets, challenging previous assumptions.

- **Pruning efficiency and data efficiency.** LTH and early-epoch pruning techniques have attracted considerable attention for their promising performance. However, the computationally intensive cycles required to identify winning tickets present a significant barrier to their wider practical deployment. While You et al. (2019) and Zhang et al. (2021b). have made strides in addressing this issue - You et al. (2019). by employing Hamming distance to track the evolution of masks across epochs, and Zhang et al. (2021b). by introducing the Pruning -Aware Critical (PrAC) set to optimize data selection - their experiments have predominantly focused on the realm of early-epoch pruning. Building upon this foundation, this study expands into the domain of KD-LTS and pruning at initialization. Findings, depicted in the plots, highlight the efficiency of KD-LTS in significantly reducing the number of training epochs required, achieving comparable results with just 70% of the typical training duration. Furthermore, we demonstrate data efficiency by successfully extracting winning tickets utilizing merely 70% of the full dataset.

- **Distillation schemes.** Ablation studies contrast offline and online distillation approaches, finding that pre-trained teacher models are not essential for identifying high-performing winning tickets. A randomly initialized teacher model and online transfer of knowledge also produce strikingly similar performance.

- **Distillation temperature.** Observations disclose that teacher models trained with varying levels of distillation temperature $T$ for soft-target response-based distillation result in distinct performances of winning tickets, aligning with findings on conventional knowledge distillation by Müller et al. (2019). Specifically, models trained with a $T$ less than 1, which form less discernible clusters at the penultimate layer activations, lead to the identification of higher-quality winning tickets. Conversely, teacher models with $T$ greater than 1, which form defined but broader clusters, correlate with a decrease in the performance of winning tickets.

- **Knowledge categories.** We explore various knowledge domains to effectively identify winning tickets, aiming to enrich the quality of our distillation framework by leveraging, diversity within ensembles through Adaptive Ensemble Knowledge Distillation (AE-KD) (Malinin et al., 2019), first-order Jacobian information (Czarnecki et al., 2017), attention maps, data diversity employing a Contrastive Representation Distillation (CRD) framework (Tian et al., 2019), (Zagoruyko and Komodakis, 2016), critical period-aware

mutual information between features Passalis et al. (2020); Park et al. (2019), layer diversity by exploiting the solution of flow procedure (FSP) (Yim et al., 2017), and, feature-based simple Mean Squared Error (MSE) loss (Ba and Caruana, 2014). A notable observation is that, although exploiting the diversity among heterogeneous features and layers indicates that improved performance winning tickets can be obtained, a simple response-based distillation with a KL-divergence loss on the logits of the student and teacher models, along with cross-entropy, is sufficient for extracting winning tickets. This study not only aids in understanding the effectiveness of KD-LTS framework but also analyses various signal telemetry to identify optimal signals for future research in developing a scoring-based function for pruning at initialization.

---

**Key Findings**

• **Commensurate reduction in complexity.** Ablation studies focusing on the number of teacher models, training phases of the teacher model, data efficiency, and pruning efficiency underscore the KD-LTS framework's contributions to computational and storage efficiency, showcasing its practicality as a real-world tool for identifying winning tickets. • **Dynamics of pruning at initialization.** Ablation studies on knowledge distillation categories and schemes not only assist in deconstructing the intricate dynamics of KD-LTS but also highlight critical signals that can systematically facilitate the development of techniques for identifying winning tickets at initialization, thereby eliminating the need to train the dense network altogether.

---

## C    Can You Win Everything with A Winning Ticket "at Initialization"?

In this paper, we meticulously evaluate the test accuracy of KD-LTS subnetworks across diverse state-of-the-art DNNs (He et al., 2016; Simonyan and Zisserman, 2014), renowned for their exceptional real-world task performance. However, we recognize the imperative need to assess neural networks beyond mere test accuracy. In today's world, where billions rely on DNN-powered technologies such as ChatGPT chatbots, known for their advanced prompt learning capabilities (Brown et al., 2020; Radford et al., 2019), a comprehensive assessment of neural network performance becomes crucial. A glimpse into key performance dimensions and their practical applications reveals, • **Interpretability:** Paramount in sectors like medical diagnosis (Gupta et al., 2021; Jiang et al., 2012), interpretability ensures model transparency and user trust by making complex models understandable. • **Confidence:** Vital in safety-critical applications such as autonomous driving (Tian et al., 2018; Bojarski et al., 2016), where the reliability of a model's predictions is directly linked to safety outcomes. • **Robustness:** Essential for security, robustness involves the network's resilience to adversarial attacks (Goodfellow et al., 2014) and noisy data, guaranteeing consistent performance under uncertain conditions, such as in cybersecurity applications Pereira and de Carvalho (2019). • **Transferability:** Critical for the versatility of neural networks, enabling models to effectively adapt across various domains with minimal retraining (Long et al., 2015), like language model adaptation across different languages or dialects Zampieri et al. (2020). • **Out-of-Distribution (OoD) Performance:** Crucial for reliability, OoD performance measures a model's ability to handle data significantly different from its training distribution, ensuring competent operation in dynamic settings (Liang et al., 2017), such as in financial modeling Ma et al. (2021b), where models must adapt to unforeseen conditions. Each of these dimensions is vital for ensuring that neural networks meet the ever evolving demands of real-world applications.

Having underscored the necessity for a holistic evaluation framework, we now meticulously analyze the performance of KD-LTS subnetworks extracted right from initialization across a wider array of critical performance dimensions, as outlined in (Chen et al., 2022; Morcos et al., 2019; Chen et al., 2021b).

### C.1    Generalization to Distribution Shifts

In the forthcoming subsections, we first define the metrics employed for evaluation, ensuring a clear understanding of our assessment criteria. We then present the principal findings, thoroughly discussing the implications and insights derived from the experiments. The following examination evaluates how KD-LTS

subnetworks fare against altered or shifted data distributions, focusing particularly on their adaptability to scenarios involving natural corruptions, adversarial perturbations, and OoD data.

**Definition C.1** (Accuracy). The accuracy of a pruned student network, denoted by $f_L^{\mathrm{s}}(x; m \odot \theta)$, and parameterized by weights $\theta$ and a binary mask $m$, is the ratio of the number of correctly predicted labels to the total number of input samples. Formally, accuracy is defined as:

$$\text{Accuracy} = \frac{1}{I} \sum_{(x,y)\in\mathbb{D}} \delta\left(\arg\max_{c\in\{1,\ldots,C\}} f_{cL}^{\mathrm{s}}(x; m\odot\theta) = \arg\max_{c\in\{1,\ldots,C\}} y_c\right)$$

where $\delta(\cdot)$ is the indicator function, which evaluates to 1 if the predicted label matches the true label and 0 otherwise. Here, $\arg\max_{c\in\{1,\ldots,C\}} f_{cL}^{\mathrm{s}}(x; m\odot\theta)$ signifies the predicted label and $\arg\max_{c\in\{1,\ldots,C\}} y_c$ denotes the ground-truth label for the input sample $x$.

The following concepts, directly related to evaluating a model's test accuracy, are integral in validating its capacity to generalize across distribution shifts:

- **Generalization Gap:** It is the discrepancy between the accuracy on the training data $\mathbb{D}_{\mathrm{train}} \subset \mathbb{D}$ and the testing data $\mathbb{D}_{\mathrm{test}} \subset \mathbb{D}$, reflecting the model's ability to generalize from the training data to data unseen over the course of training.

- **Natural Corruption Robustness:** The mean corruption error (mCE) assesses the model's robustness to various natural corruptions and is defined as:

$$\mathrm{mCE} = \frac{1}{N_{\mathbb{C}}} \sum_{\kappa=1}^{N_{\mathbb{C}}} \left[\frac{\sum_{\rho=1}^{5} E_{\rho,\kappa}^{m\odot\theta}}{\sum_{\rho=1}^{5} E_{\rho,\kappa}^{\theta}}\right].$$

Here, $N_{\mathbb{C}}$ represents the total number of corruption types considered, $\kappa$ is used as the variable to iterate over the corruption types. The metric leverages the top-1 error rates $E_{\rho}^{m\odot\theta}$ and $E_{\rho}^{\theta}$ for the sparse and dense models, respectively, across a range of corruption types such as noise, blur, weather, digital process applied across input samples $x$. Each corruption type has five corruption severity levels (i.e., $1 \leq \rho \leq 5$). CIFAR-10/100-C from (Hendrycks and Dietterich, 2019) are adopted in our experiments for corrupted data.

  **Type of natural corruptions.** (Hendrycks and Dietterich, 2019) provides 19 natural corruptions: "gaussian noise", "shot noise", "impulse noise", "defocus blur", "glass blur", "motion blur", "zoom blur", "snow", "frost", "fog", "brightness", "contrast", "elastic transform", "pixelate", "jpeg compression", "speckle noise", "gaussian blur", "spatter", "saturate". More information can be found in the official repository `https://github.com/hendrycks/robustness/tree/master/ImageNet-C/imagenet_c`.

- **Adversarial Robustness:** The model's robustness against adversarial attacks is gauged by its accuracy on adversarially perturbed images. These perturbations for input samples $x$ are crafted using a targeted Projected Gradient Descent (PGD) attack (Madry et al., 2017), formulated as:

$$x_{n+1} = \Pi_{x+\upsilon}\left(x_n + \epsilon_{\mathrm{adv}}\mathrm{sgn}\left(\nabla_{x_n} L_{\mathrm{CE}}\left(x_n; (m\odot\theta)\right)\right)\right)$$

where $\epsilon_{\mathrm{adv}}$ is the magnitude of the perturbations, $\Pi$ is a projection operator, $\upsilon \in \mathcal{R}^d$ represents the set of perturbations illustrating the model's stability under adversarial conditions. The epsilon value ($\epsilon$) and the number of iterations $i$ employed in this iterative targeted PGD attack are detailed alongside the experimental plots.

- **Out-of-Distribution (OoD) Performance:** The model's ability to make reliable predictions on OoD data is assessed using the ROC-AUC metric. ROC-AUC stands for the area under the receiver operating characteristic (ROC) curve. This evaluation is crucial for comprehending the model's predictive confidence when confronted with data that exhibits statistical characteristics divergent from those present in the training set. Such an understanding is indispensable to ensure reliability in real-world applications where data distribution may constantly change. Following (Hendrycks and Gimpel, 2016; Krizhevsky et al., 2009), for CIFAR-10 experiments, CIFAR-100 is regarded as the OoD dataset; for CIFAR-100 experiments, CIFAR-10 is selected as the OoD dataset.

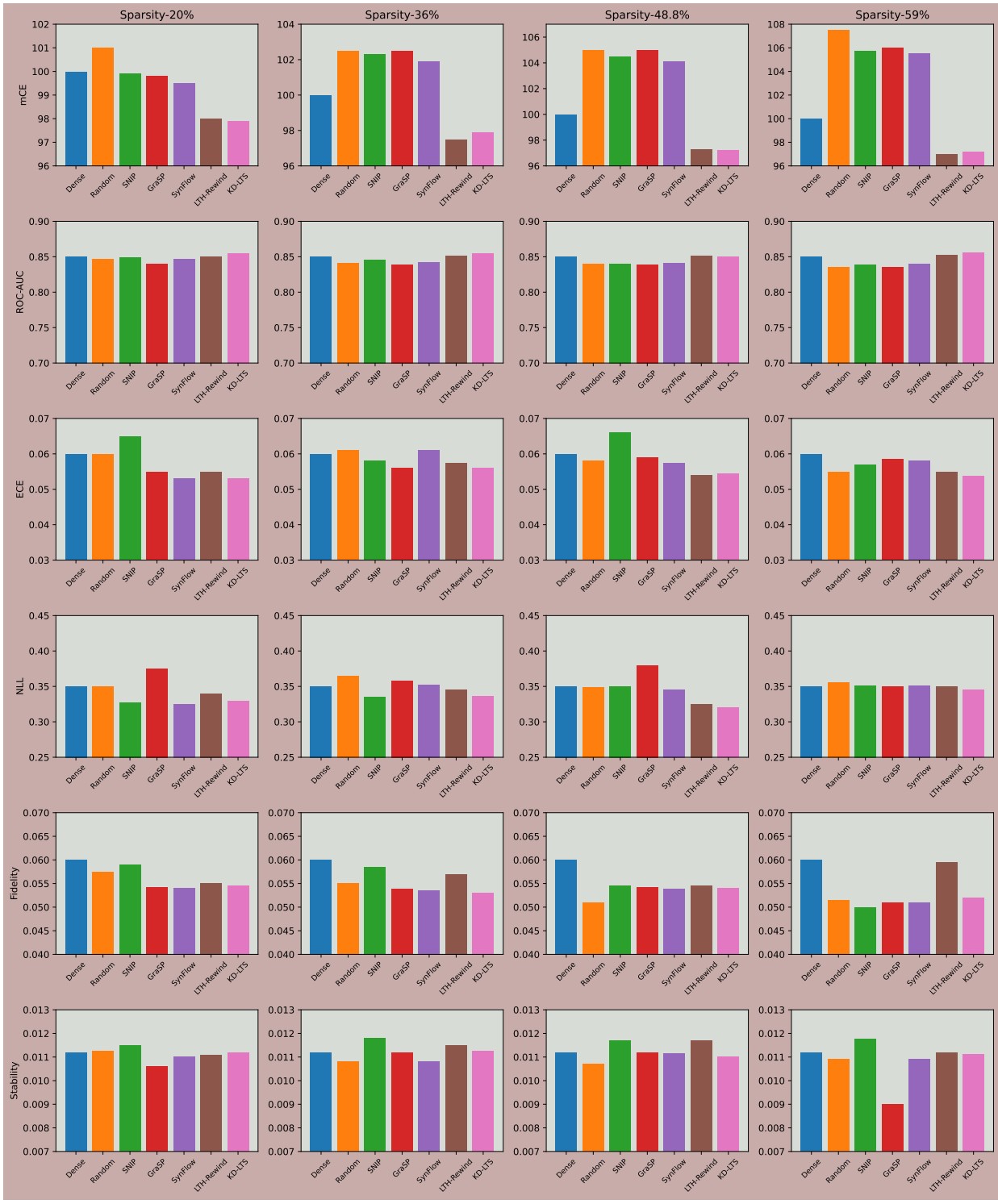

Figure 16: Juxtaposition of KD-LTS subnetworks on metrics such as Natural Corruption Robustness (mCE), Out-of-Distribution (OoD) performance (ROC-AUC), Expected Calibration Error, Negative Log Likelihood, Fidelity, and Stability, against early pruning, initialization pruning, and dense networks in a ResNet-20/CIFAR-10 environment.

The results related to natural corruption robustness and out-of-distribution (OoD) performance are presented in Figure 16, while the findings on adversarial robustness are comprehensively detailed in Figure 17. It is well-established that sparse networks typically demonstrate superior generalization to distribution shifts, a characteristic often linked to their improved Lipschitz interpretability. Therefore, in our experiments designed to showcase the KD-LTS subnetworks' generalization to distribution shifts, we extend our comparisons beyond dense networks to include sparse networks extracted through existing early and initialization pruning methods. This extension offers a comprehensive understanding and thorough evaluation of their competitive performance.

> **Key Findings**
>
> • **Robustness.** KD-LTS subnetworks demonstrate a significant 3-4% improvement in Mean Corruption Error (mCE), a 1-2% increase in ROC-AUC, and a 2-3% enhancement in adversarial robustness compared to subnetworks derived through pruning at initialization. Furthermore, they attain performance comparable to subnetworks extracted by early pruning methods, highlighting KD-LTS's superior efficacy across various sparsity levels.

**How does KD-LTS assert generalization to distribution shifts?** Figure 17 showcases sample classifications of KD-LTS winning tickets on adversarial examples, demonstrating how misclassification only occurs when the features of the sample appear closer to the adversarial target class compared to random subnetworks and the dense network. Figures 23, 24, and 25 delineate the robust decision-making capabilities of KD-LTS subnetworks against random subnetworks and the dense network in a LeNet-300-100/MNIST experiment, as examined through the lens of the LIME explainer (Ribeiro et al., 2016). This serves to substantiate their inherent robustness. Figures 26 and 18 focus on the activations of the penultimate layer in KD-LTS, dense, and random subnetworks. In KD-LTS subnetworks, our experiments reveal that projections form well-defined, tight clusters indicating clear class boundaries. In contrast, for dense and random subnetworks, these projections appear more scattered and less discernible, highlighting the superior adaptability of KD-LTS subnetworks to distribution shifts.

## C.2  Calibration and Reliability

Confidence calibration (Guo et al., 2017; Venkatesh et al., 2020) is of paramount importance to ensure reliability in situations where incorrect predictions could lead to serious consequences. To evaluate the calibration of confidence in the predictions made by KD-LTS subnetworks, we adopt the following standard metrics:

**Definition C.2** (Confidence)**.** Let $f_L^s(x; m \odot \theta)$ denote the prediction of a pruned student model for a given input $x$ from a dataset $\mathbb{D}$. The confidence of this prediction is defined as the predicted probability that the model assigns to the most likely class. Formally, for a dataset $\mathbb{D}$, the average confidence over the set of predictions is given by:

$$\text{Confidence} = \frac{1}{I} \sum_{x \in \mathbb{D}} \frac{e^{f_{c^* L}^s(x; m \odot \theta)}}{\sum_{c=1}^{C} e^{f_{cL}^s(x; m \odot \theta)}},$$

where $c^*$ represents the class with the highest probability predicted by the model for the input $x$.

Leveraging both accuracy and confidence, we present explanations of the following metrics associated with confidence calibration.

- **Expected Calibration Error (ECE)**: ECE (Naeini et al., 2015) is defined as the weighted average of the differences between accuracy and confidence across $Z$ bins:

$$\text{ECE} = \sum_{z=1}^{Z} \frac{N_{B_z}}{I} \left| \text{Accuracy}(B_z) - \text{Confidence}(B_z) \right|,$$

where $I$ represents the number of samples in the dataset $\mathbb{D}$ and $N_{B_z}$ is the number of samples in the $z$-th bin. This metric serves as a proxy for the mismatch between the model's confidence and its actual performance.

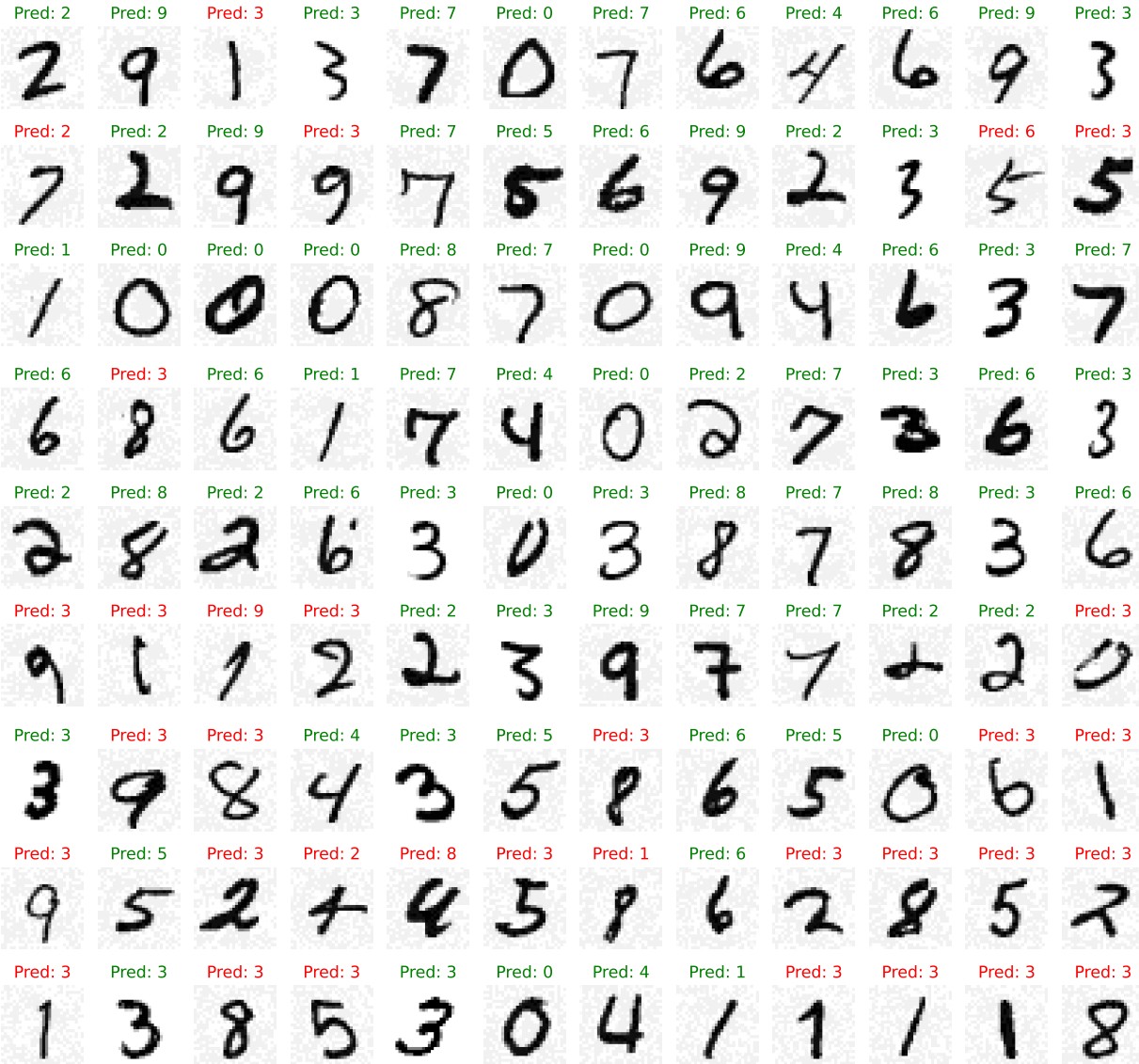

Figure 17: Sample classification of adversarial examples in an MNIST data perturbed using a targetted PGD attack (target label '3') by a dense LeNet-300-100, KD-LTS extracted 45% sparse winning ticket, and 45% sparse randomly pruned subnetwork.

- **Negative Log Likelihood (NLL)**: NLL is a measure of how well the log probability distribution outputted by the model, aligns with the true labels:

$$\text{NLL} = -\sum_{(x,y)\in\mathbb{D}_{\text{val}}} y^{\mathsf{T}}\log\left(\mathcal{P}(z|x)\right),$$

where $\mathbb{D}_{\text{val}}$ is a validation subset of the dataset $\mathbb{D}$. NLL (LeCun et al., 2015) is minimized when the predicted distribution converges to the true distribution of the labels, indicating optimal calibration.

The results for ECE and NLL in the ResNet-20 experiment on CIFAR-10 are illustrated in Figure 16. Similarly, the outcomes of these metrics for the CIFAR-100 experiment are depicted in Figure 30.

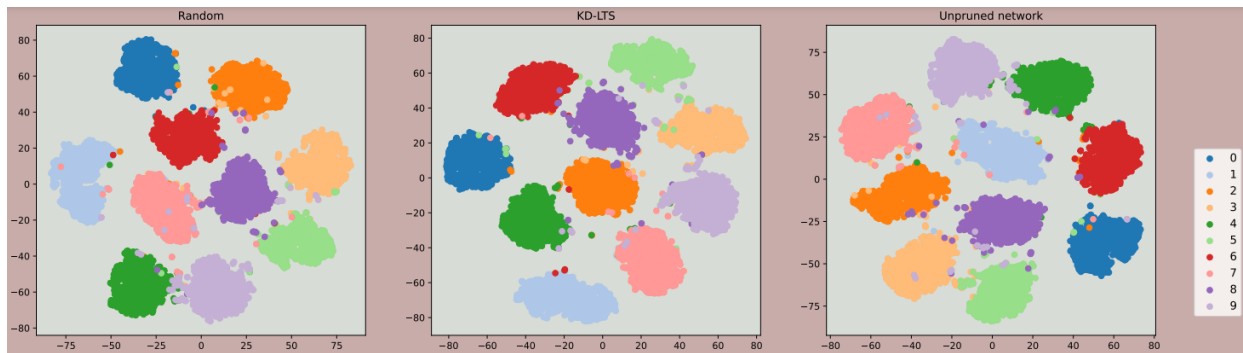

Figure 18: Visualization of penultimate layer's activations across the ten classes in KD-LTS subnetwork, random subnetwork, and unpruned VGG-16 network trained on CIFAR-10.

> **Key Findings**
>
> • **Reliability.** KD-LTS subnetworks demonstrate the ability to maintain or enhance confidence levels compared to dense and early-pruned subnetworks, across a range from denser 20% to rarer 60% sparsity levels. This attribute renders KD-LTS subnetworks viable for reliable deployments in real-world settings.

### C.3 Interpretability

We investigate the capability of winning tickets identified by KD-LTS to preserve interpretability. The assessment involves a quantitative analysis of explainability, grounded in functional representation, employing the most widely-recognized metrics: namely, *fidelity* (Plumb et al., 2020) and *stability* (Ghorbani et al., 2019).

- **Fidelity**: Given a validation dataset $D_{\text{val}}$, fidelity quantifies the extent to which the local explanations of a model's predictions align with the actual outcomes of the model. Formally, it is defined as the expected difference in the model's predictions for the original input and a perturbed input within a defined neighborhood, and is given by:

$$\text{Fidelity} = \mathbb{E}_{x \in D_{\text{val}}} \left[ \mathbb{E}_{x' \sim \mathcal{N}_x} \left[ |g(x') - f(x')|^2 \right] \right],$$

  where $g(x')$ represents the explanation of model's prediction, and $f(x')$ denotes the target model's prediction for the perturbed input $x'$.

- **Stability**: Stability $S$ focuses on the consistency of local explanations under input perturbations, reflecting the robustness of the explanation model. It is defined as:

$$\text{Stability} = \mathbb{E}_{x \in D_{\text{val}}} \left[ \mathbb{E}_{x' \sim \mathcal{N}_x} \left[ \|\mathbf{e}(x, f) - \mathbf{e}(x', f)\|_2^2 \right] \right],$$

  where $\mathbf{e}(x, f)$ and $\mathbf{e}(x', f)$ are the embeddings of the explanations for the original and perturbed inputs, respectively.

Following Chen et al. (2022), to compute these metrics, we first perturb each input image $x$ and build its neighborhood set $\mathcal{N}_x$ with a size of $1,000$ samples. Then, we generate a class of interpretable functions $G = \{g_x \in G | x \in D_{\text{val}}\}$, where $g_x$ is a linear function obtained from a regression to the corresponding model's output on $\mathcal{N}_x$. In the above formulation, $f(\cdot)$ denotes the target model we want to interpret, $\mathbf{e}(x, f)$ and $\mathbf{e}(x', f)$ are the learned weights of linear models $g_x$ and $\tilde{g}_x$. Each training sample $\tilde{x} \in \mathcal{N}_x$ is weighted by the Hamming distance of $(\tilde{x}, x)$ and $(\tilde{x}, x')$, respectively. All our experiments replicate the implementation detailed in (Ribeiro et al., 2016). Intuitively, fidelity quantifies how accurately the explainer $g_x$ models the target network $f$ in a neighborhood $\mathcal{N}_x$; Stability measures the degree to which the explanation changes across points in $\mathcal{N}_x$.

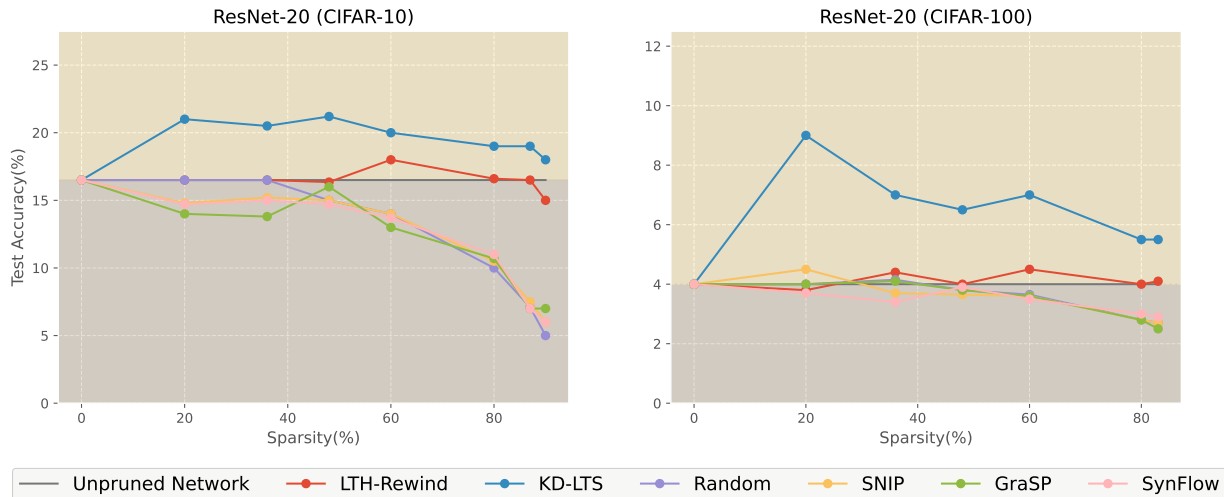

Figure 19: Adversarial accuracy of winning tickets obtained in a ResNet-20 network across datasets CIFAR-10 (left) & CIFAR-100 (right).

The stability and fidelity results from the ResNet-20/CIFAR-10 experiment are presented in Figure 16, while the corresponding results for the CIFAR-100 experiment are shown in Figure 30

> **Key Findings**
>
> • **Stability and precision.** KD-LTS winning tickets showcase a 2-5% improvement in stability and fidelity compared to LTH-Rewind, reaffirming our claim that KD-LTS is not just an alternative but a viable replacement for early pruning techniques, ensuring both software and hardware cost reduction.

## C.4 Transferability

We expand on the conjecture proposed in (Morcos et al., 2019; Chen et al., 2021b) regarding the transferability of winning tickets across various datasets, optimizers, and architectures. While LTH-Rewind has been the dominant method for identifying transferable winning ticket, our research re-evaluates and broadens this concept to extract transferable subnetowrks right at the initialization stage. Our empirical findings clearly demonstrate that winning tickets identified through the KD-LTS framework are indeed transferable across different datasets, optimizers, and architectures. Notably, they even exhibit performance that is comparable, and in most instances, superior to those identified by LTH-Rewind experiments.

> **Key Findings**
>
> • **Versatility.** KD-LTS subnetworks show an improvement of 0.6%, 0.7%, and 0.42% in performance when transferred across datasets (CIFAR-10 ← CIFAR-100 and CIFAR-100 ← CIFAR-10), optimizers (Adam ← SGD), and architectures (ResNet-14 ← ResNet-20), respectively, compared to LTH-Rewind transferability. • **Universality.** The results underscore the existence of shared and transferable patterns in DNNs that make sparse networks trainable, not just from early epochs but right from the initialization stage. Furthermore, they demonstrate the KD-LTS framework's capability in identifying these patterns.

In summary, KD-LTS subnetworks, extracted right from the onset of training, not only enjoy superior test performance but also provide sparse networks that are accurate, transferable, interpretable, reliable, secure, and confident.

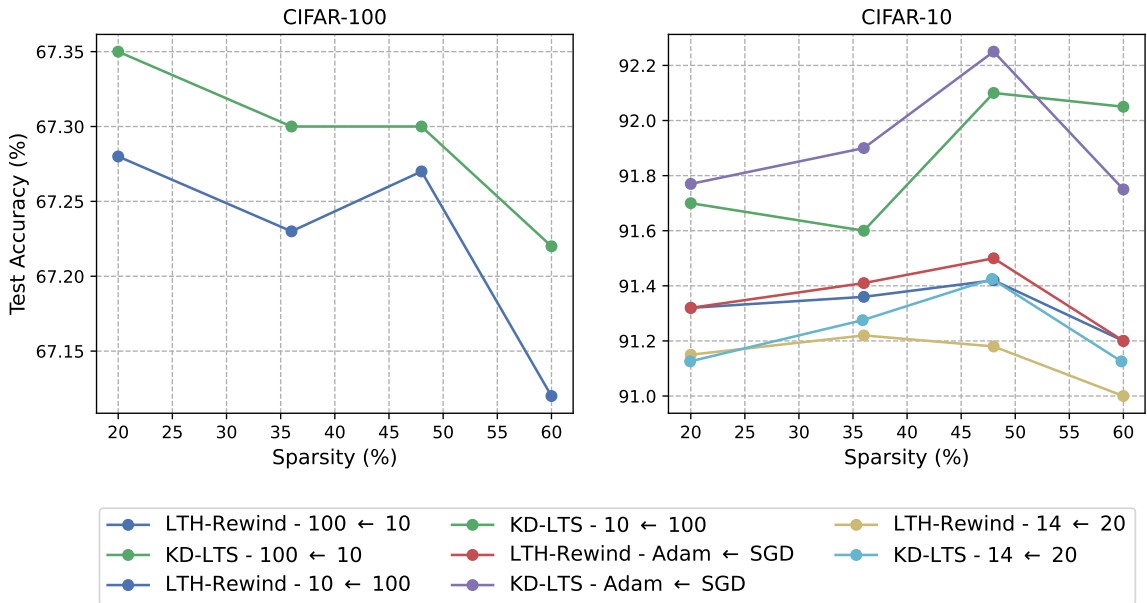

Figure 20: Test accuracy of winning tickets transferred across datasets (CIFAR-10 & CIFAR-100), architectures (ResNet-20 & ResNet-14) and Optimizers (Adam & SGD).

## D  Supermasks

**Definition D.1** (Supermask)**.** A supermask is a subnetwork, derived from a randomly initialized neural network, represented as $f_L(x; m \odot \theta^{(0)})$, capable of achieving a test accuracy surpassing the chance accuracy without the need for training the underlying parameters. Here, the term chance accuracy refers to the test accuracy achieved by the original, untrained dense network, denoted as $f_L(x; \theta^{(0)})$.

Recent theoretical studies (Malach et al., 2020; Orseau et al., 2020; Ramanujan et al., 2020) have substantially strengthened the proposal of LTH. These works suggest that a sufficiently overparameterized neural network contains a subnetwork that, with only logarithmic increase in size compared to the target network, can achieve comparable accuracy to the large network - even without any training. In response to these groundbreaking insights on LTH, a considerable amount of experimental research has been conducted. Efforts include CS (Savarese et al., 2019), Stochastic Sparsification (SS) (Zhou et al., 2019), Edge Popup (EP), Smart Ratio (Su et al., 2020), and GEM-MINER (Sreenivasan et al., 2022), all aimed at developing algorithms for identifying *supermasks*. This burgeoning area of research underscores the latent potential of subnetworks at initialization.

These theoretical results indicate that pruning alone might suffice as a training method. This challenges the conventional belief that binary mask training, perceived as having lesser representational power compared to continuous parameter training, cannot yield comparable outcomes. Supermasks confer a notable advantage in terms of storage efficiency, necessitating only the binary mask and a single random seed for the reconstruction of the entire network's weights. KD-LTS further strengthens this characteristic, obtaining high-quality supermasks that can be harnessed for their own line-up of applications in real-world scenarios.

**Exploration into a novel avenue of knowledge distillation.** KD-LTS paves the way for various avenues of exploration, extending beyond pruning to include a pioneering premise in knowledge distillation. Unlike traditional knowledge distillation methods that transfer knowledge from larger teacher models to smaller student networks via weight training, KD-LTS involves mask-training large student networks using heterogeneous information from teacher networks. The pruned larger networks, known as supermasks, exhibit exceptional standalone performance without any supplementary training. These supermasks are not only receptive to further weight-training from scratch but also exceed the accuracy levels and parameter efficiency of dense, small student models trained through conventional knowledge distillation methods. For example, consider the case of ResNet-50, with its approximately 24 million parameters. Utilizing KD-LTS for pruning

at initialization, we obtain a subnetwork with 90% sparsity that exceeds the performance of a fully-trained ResNet-18, which has around 12 million parameters and is trained using standard knowledge distillation techniques. This achievement is particularly remarkable, considering that the KD-LTS-identified subnetwork benefits from an 80% reduction in parameter count. Notably, our knowledge distillation framework highlights that these winning tickets not only achieve a notable 3.4% improvement in test accuracy but also demonstrate that supermasks, even without any additional training, experience only a marginal 3% reduction in accuracy compared to their traditional knowledge distillation counterparts.

---

**Key Findings**

• **Test efficiency.** Among the comparative techniques, IMP and SR produce supermasks with accuracy levels barely surpassing random guessing, SS struggles to learn masks with sparsity exceeding 50%, and EP identifies subnetworks with high pre-training accuracy but faces limitations in the high sparsity regime, particularly beyond 95%. In stark contrast, KD-LTS method stands out, with its highest-performing supermasks achieving 94%, 66.4%, 90.1%, and 74.4% test accuracy at 98% sparsity for LeNet-300-100, ResNet-20, VGG-16, and a 6-layer CNN, respectively. These results significantly surpass the leading results presented in the current literature using other techniques, which are 60%, 61.5%, 88.2%, and 68.2%, establishing a marked improvement over existing methods. • **Unexplored arena.** KD-LTS not only pioneers new avenues in the field of neural network pruning at initialization but also offers a novel perspective on knowledge distillation, significantly contributing to the realm of knowledge distillation-assisted model compression.

---

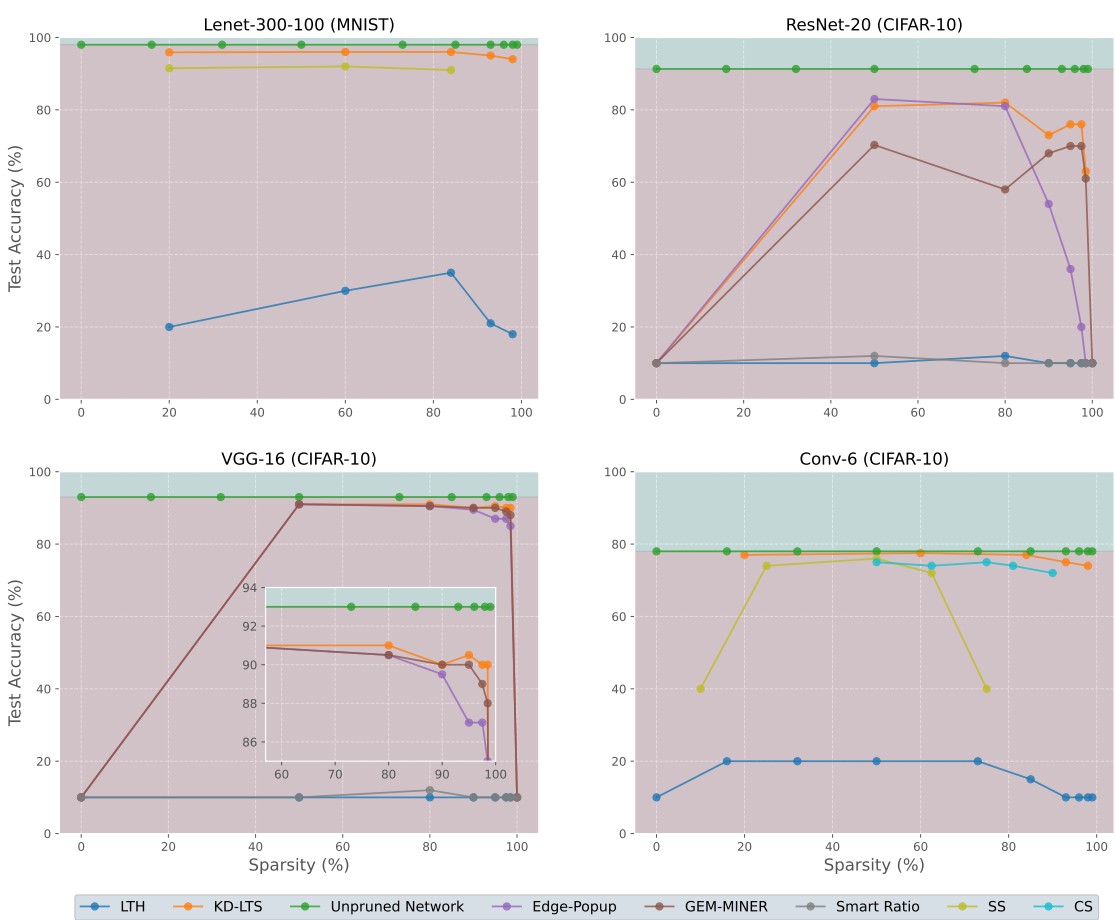

Figure 21: Evaluating the supermask performance of KD-LTS in comparison to CS, SS, GEM-MINER, Edge-Popup, Smart Ratio, and LTH.

# E Future Work

Acknowledging room for improvement within KD-LTS, the key areas for our future investigation include: • Extend KD-LTS beyond computer vision to diverse domains such as natural language processing. • Investigate the transferability of winning tickets across applications for broader impact, surpassing the current focus on optimizers, networks, and datasets. • Integrate KD-LTS to identify subnetworks specifically optimized for modern libraries and hardware, achieving tangible computational reductions. • Examine the structure, patterns, and characteristics of KD-LTS identified subnetworks to investigate their efficacy and potential as a powerful initialization technique. (**Intriguing Observation:** We observed that KD-LTS, by transferring information on the SVD of parameters (KD+SVD+CE), identified winning tickets achieving an adversarial accuracy of 79% after training. This contrasts with subnetworks identified using cross-entropy and label information alone, which achieved 63% adversarial accuracy. Importantly, both sets of subnetworks underwent only standard training with standard data. This demonstrates the surprising capability of a pruning-at-initialization technique like KD-LTS to embed specialized traits, such as adversarial accuracy, using only standard data - traits previously thought to be achievable solely through computationally intensive training methods, such as adversarial training with adversarial data). • Develop effective techniques for efficient sparsity control within the Phase-3 of KD-LTS.

## F  Additional Experiments

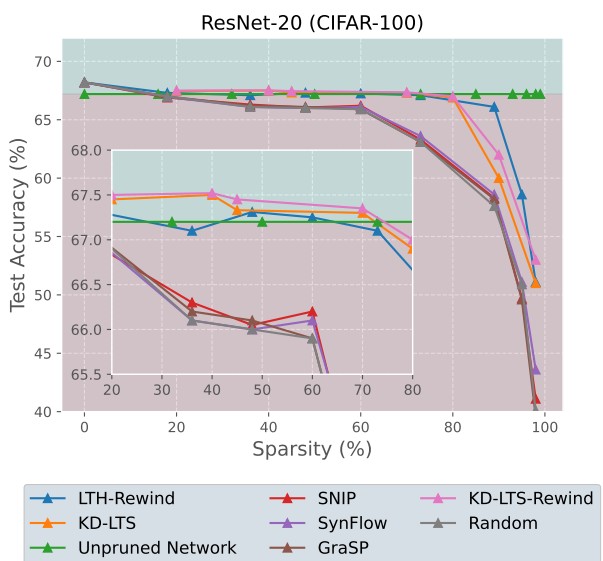

Figure 22: KD-LTS' winning ticket performance, extracted at initialization and early epoch.

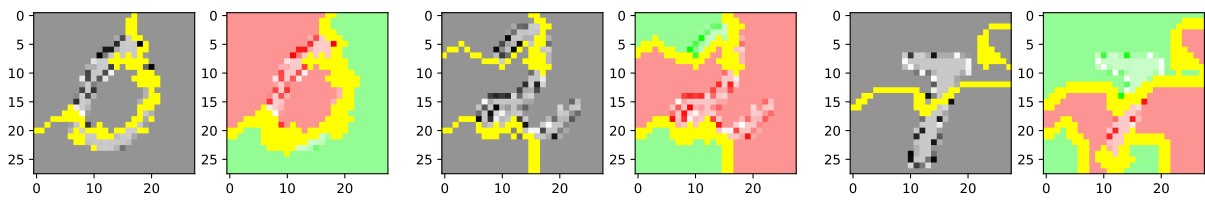

Figure 23: Highlighting the positive/negative regions for classifications of MNIST data by trained KD-LTS subnetwork, unpruned network and a random subnetwork in a LeNet-300-100.

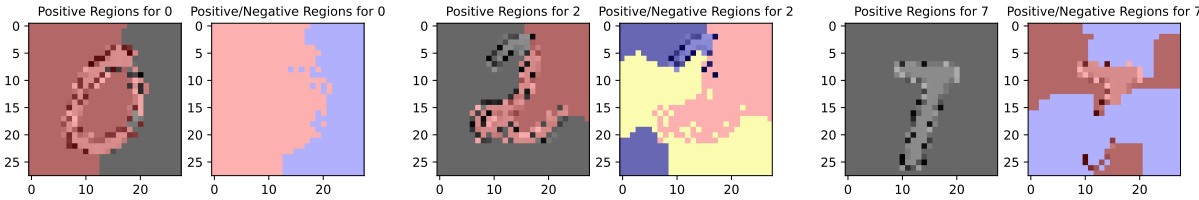

Figure 24: Explaining the positive/negative regions for classifications of MNIST data by trained KD-LTS subnetwork, unpruned network and a random subnetwork in a LeNet-300-100.

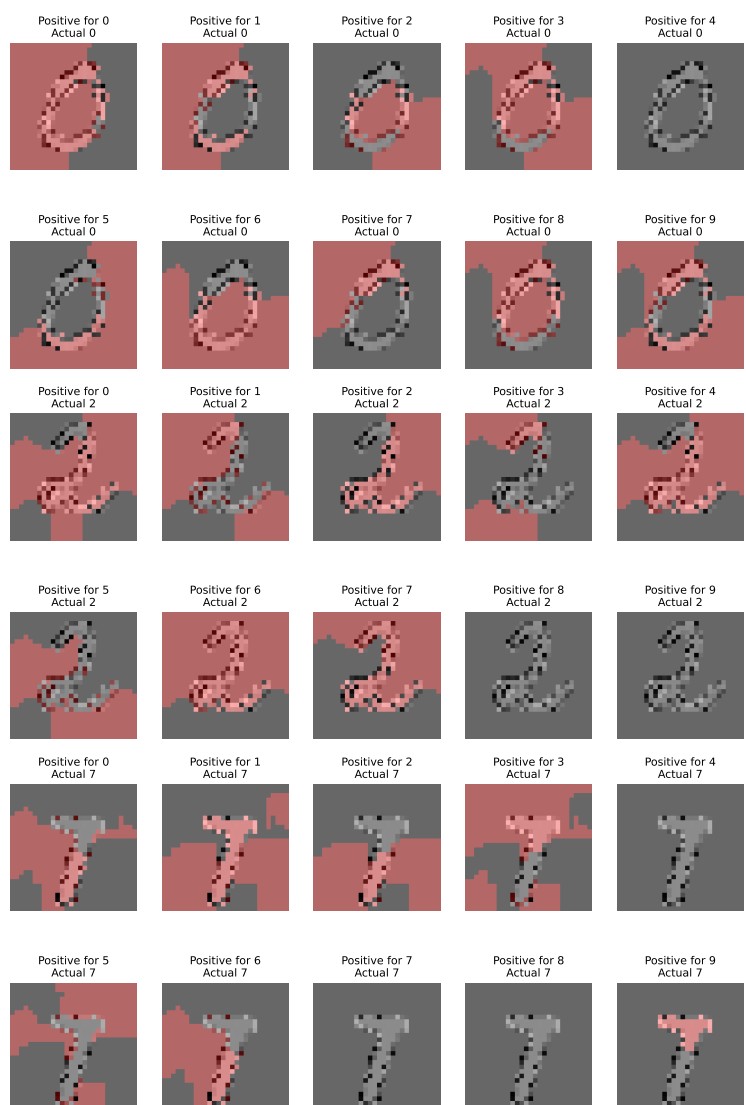

Figure 25: Explaining the classifier: A KD-LTS subnetwork, unpruned network, and a random subnetwork in a LeNet-300-100 in terms of positive regions associated with classification of a ground truth label against positive regions for misclassification.

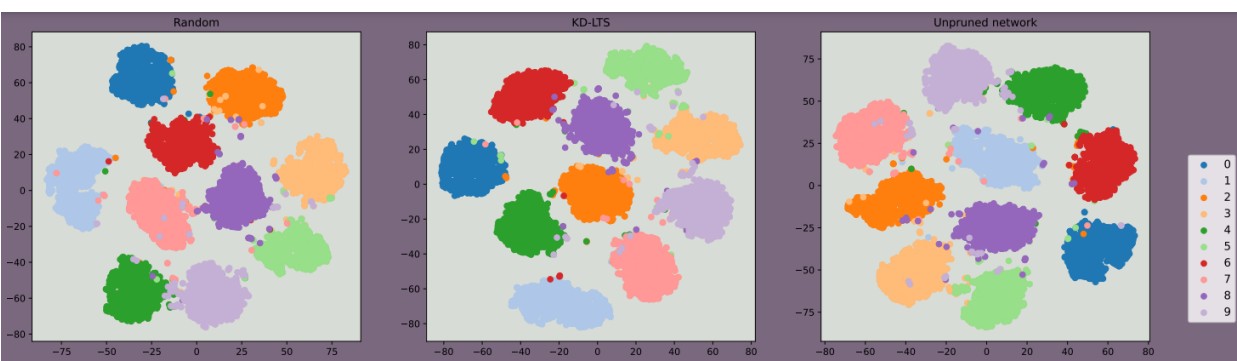

Figure 26: Visualization of penultimate representations in KD-LTS subnetwork, Random subnetwork, and Unpruned VGG-16 network trained on CIFAR-10.

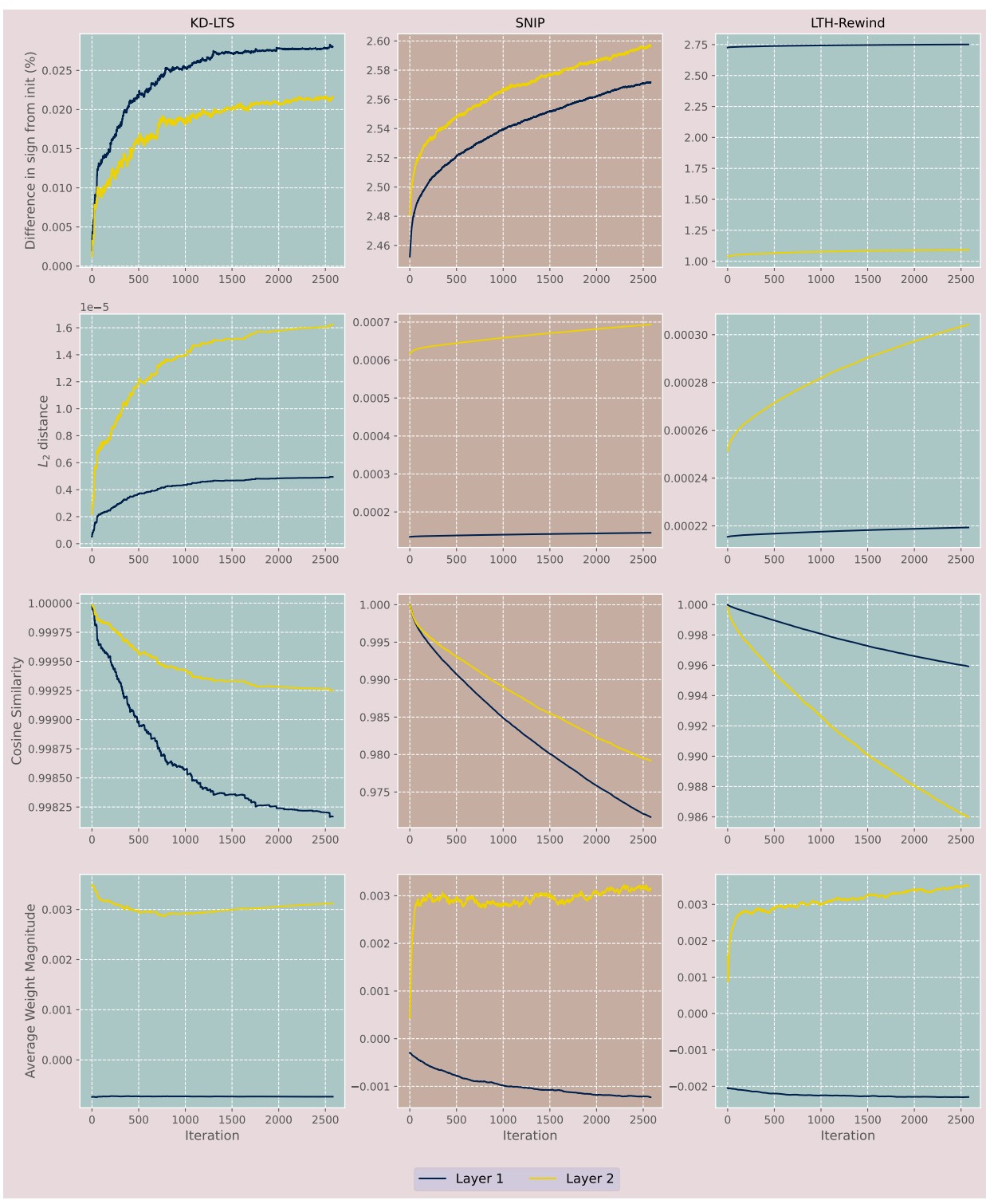

Figure 27: Training dynamics of weight parameters analyzed through the lens of pivotal metrices in a LeNet-300-100 network across the first 2580 iterations.

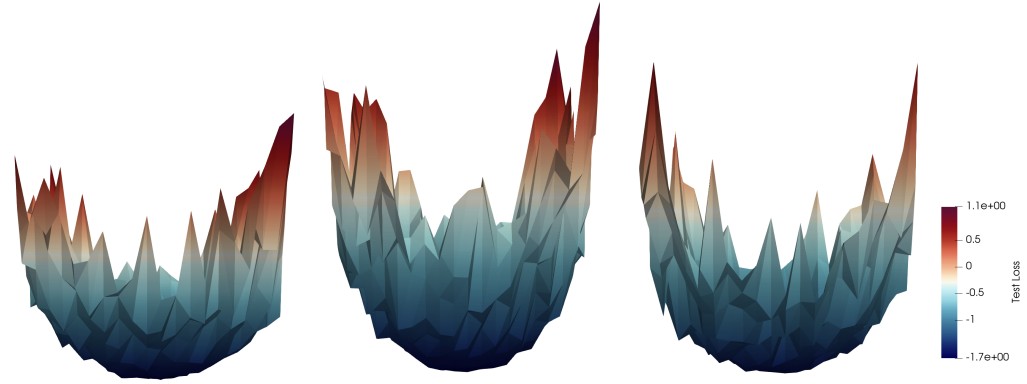

Figure 28: Analyzing the optimization landscape in LeNet-300-100 subnetwork trained on MNIST over the first six epochs.

Figure 29: Visualizing the Loss landscape: Comparing LeNet-300-100 subnetworks at the optimum extracted via KD-LTS, Random, and LTH-Rewind, trained on the MNIST dataset

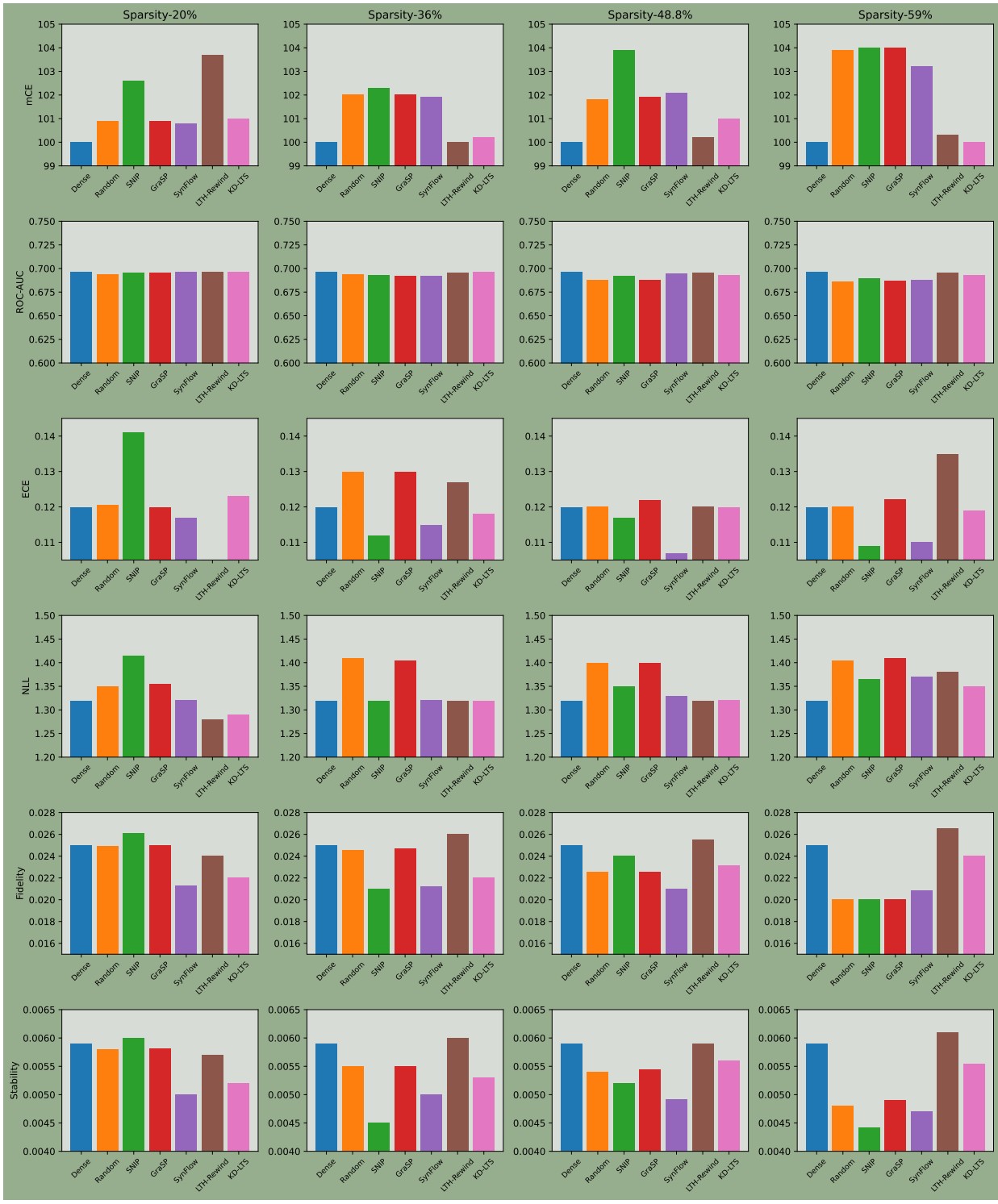

Figure 30: Comparative analysis of KD-LTS subnetworks on Key Metrics: Natural Corruption Robustness (mCE), Out-of-Distribution performance (ROC-AUC), Expected Calibration Error, Negative Log Likelihood, Fidelity, and Stability against early pruning, initialization Pruning, and dense networks in a ResNet-20/CIFAR-100 environment.

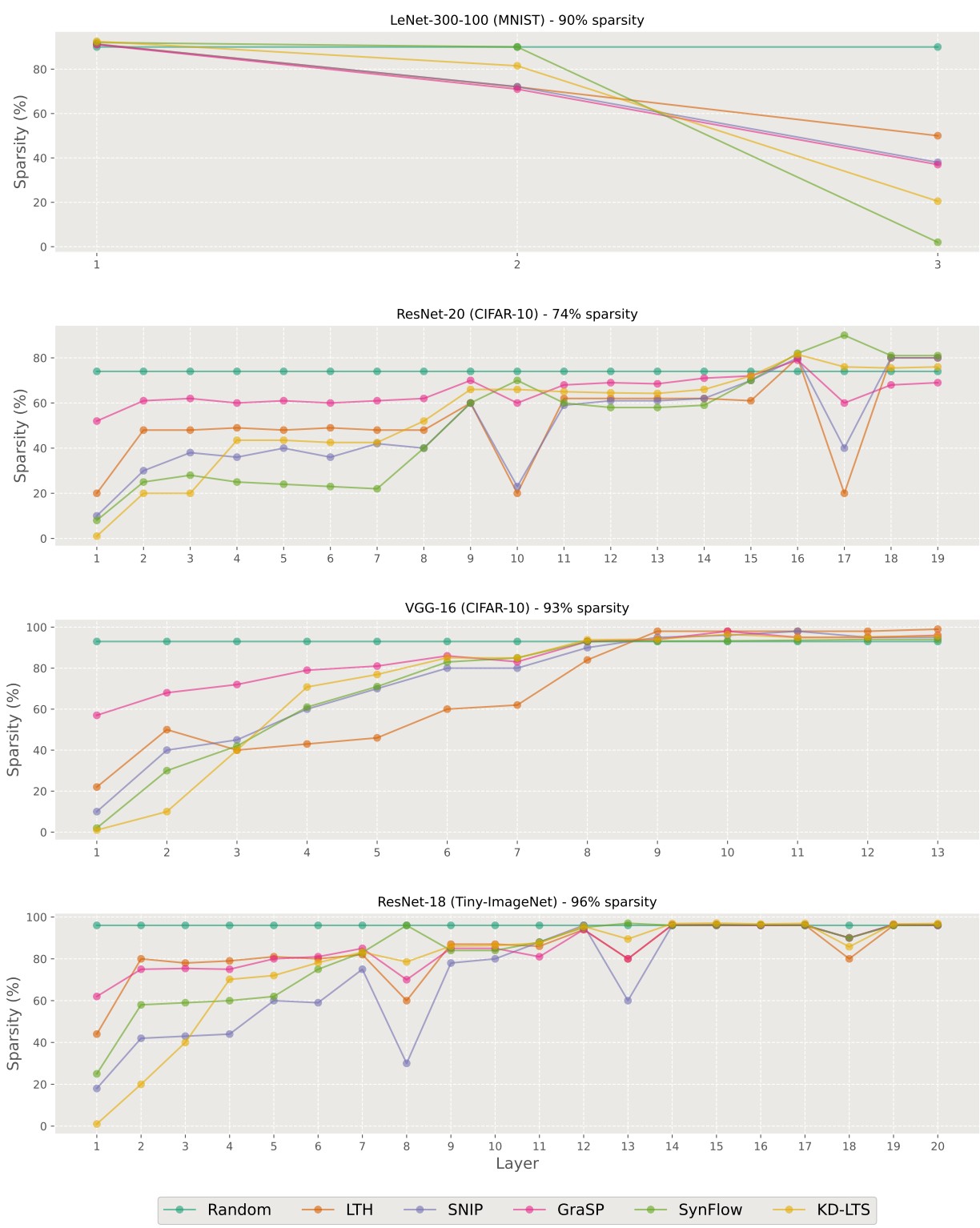

Figure 31: Per-layer sparsities achieved by each pruning at initialization method for a specified sparsity level across various experiments.

# G   Experimental Setup

This section provides a detailed description of the experimental settings and configurations utilized to evaluate the effectiveness of proposed KD-LTS technique in detecting winning tickets at initialization.

## G.1   Datasets

Experiments are conducted on four well-established datasets, each possessing unique characteristics, to comprehensively evaluate winning tickets. For the MNIST database, 55,000 examples were utilized for training and 5,000 were reserved for validation, with images normalized on a per-channel basis. In the case of CIFAR-10 and CIFAR-100 datasets, per-channel normalization, random horizontal flipping, and random pixel shifts of up to four pixels in any direction were applied. The training sets were partitioned into 45,000 training examples and 5,000 validation examples. The model demonstrating the highest validation accuracy was chosen for further evaluation on the test set. For the Tiny-ImageNet dataset, augmentation included per-channel normalization, selection of a patch with a random aspect ratio between 0.8 and 1.25 and a random scale between 0.1 and 1, cropping to 224x224, and applying random horizontal flipping.

## G.2   Network Architectures

Four distinct neural network models were deployed, carefully chosen to underscore the versatility and robustness of the approach. This selection includes architectures suitable for both small-scale and large-scale settings, aligning with classifications from the work of Frankle et al. (2020a) and further informed by our research considerations. The first model, LeNet-300-100, is characterized by its 266K parameters. Progressing to a larger scale, VGG-16, featuring 14.7M parameters, is specifically utilized in the CIFAR-10/CIFAR-100 experiments. ResNet-20, introduced for the CIFAR-10/100 datasets with 20 layers and 274K parameters in (He et al., 2016), adds diversity to the selection. Lastly, ResNet-50, tailored for the ImageNet dataset with 50 layers and 25.5M parameters, further emphasizes the range of architectures in the experimental framework. The implementations of ResNet-20 and ResNet-50 were sourced from the OpenLTH repository, while the LeNet-300-100 and VGG-16 networks were leveraged from the TorchVision implementations.

## G.3   Training Settings

Table 4 offers a comprehensive overview of the training settings used in experiments to assess both KD-LTS and other comparative techniques for extracting subnetworks, both at initialization and from early epochs.

Table 4: Training Settings

| Network | Dataset | Epochs | Batchsize | Iterations per epoch | Rewind Iteration |
|---|---|---|---|---|---|
| LeNet-300-100 | MNIST | 40 | 256 | 469 | 1407 |
| ResNet-20 | CIFAR-10/100 | 180 | 128 | 391 | 1173 |
| VGG-16 | CIFAR-10/100 | 180 | 128 | 391 | 1173 |

Tables 5 and 6 detail the optimizer settings for the post-extraction training of subnetworks and mask training, respectively.

Table 5: Optimizer settings for training weight and bias parameters

| Network | Batchsize | Optimizer | Momentum | LR | LR Drop | Weight decay | Initialization |
|---|---|---|---|---|---|---|---|
| LeNet-300-100 | MNIST | SGD | - | 0.01 | - | - | Orthogonal |
| ResNet-20 | CIFAR-10/100 | SGD | 0.9 | 0.1 | 10x at 90, 135 | 2e-4 | Orthogonal |
| VGG-16 | CIFAR-10/100 | SGD | 0.9 | 0.1 | 10x at 80, 120 | 2e-4 | Orthogonal |

Table 6: Optimizer settings for training mask parameters

| Network | Dataset | Optimizer | Momentum | Max LR | Base LR | Weight decay | Initialization |
|---|---|---|---|---|---|---|---|
| LeNet-300-100 | MNIST | SGD | - | 0.1 | 0.001 | - | Orthogonal |
| ResNet-20 | CIFAR-10/100 | SGD | 0.9 | 0.1 | 0.001 | - | Orthogonal |
| VGG-16 | CIFAR-10/100 | SGD | 0.9 | 0.1 | 0.001 | - | Orthogonal |

Next, we present the hyperparameters for our knowledge distillation framework in Table 7. These are the general settings used across all our experiments unless otherwise stated. The hyperparameters for response-based, feature-based, and relational-based knowledge distillation techniques mentioned in the ablation studies are specified in the respective sections.

Table 7: Hyperparameters for the knowledge distillation framework

| Network | Dataset | $\alpha$ | $\beta^{(0)}$ | $k$ | $\nu_k$ | $T$ |
|---|---|---|---|---|---|---|
| LeNet-300-100 | MNIST | 0.9 | 1 | 3 | 0.34 | 0.6 |
| ResNet-20 | CIFAR-10/100 | 0.9 | 1 | 3 | 0.34 | 2 |
| VGG-16 | CIFAR-10/100 | 0.9 | 1 | 3 | 0.34 | 30 |

# H   Benchmarking Sparsification Techniques: A Comparative Study

We conduct a comprehensive evaluation of multiple baseline methods and various state-of-the-art sparsification techniques to demonstrate the advantages of our approach in extracting supermasks, conducting pruning at initialization, and performing early training pruning. Additionally, we carry out a series of sanity checks, sensitivity analyses, and stability assessments across the strategies to distinguish the quality of our winning tickets. In this section, we briefly describe the key features of these juxtaposed techniques and the critical parameters in their implementation.

**Lottery Ticket Hypothesis**: Frankle and Carbin (2018) adopted and customize the popular IMP to an iterative process to obtain subnetworks at initialization. We refer to the standard LTH procedure that involves resetting the weights to their random initial values, as 'LTH' in our plots. As an alternative, when the weights are rewound to an earlier epoch, we denote it as 'LTH-Rewind'. Throughout all LTH-Rewind experiments presented in this study, we rewind the weights specifically to the 3rd epoch (unless otherwise specified). For our experiments, the LTH implementation from the OpenLTH repository is employed. Also, to ensure the functionality of the pruned subnetwork, its weights are proportionally scaled to account for the fraction of weights pruned, as described in Algorithm 2.

---

**Algorithm 2:** LTH

---

1: Randomly initialize a neural network $f_L^s(x; (m \odot \theta^{(0)}))$ where $m = \mathbf{1}$: $m$ is a vector of all-ones with the same dimension as $\theta^{(0)}$.

2: Train the network for $E$ epochs with an optimal learning rate to minimize $L_{\text{CE}}$ and obtain $f_L^s(x; (m \odot \theta^{(E)}))$.

3: Prune 20% of the parameters with the least absolute magnitude $|\theta^{(E)}|$, creating an updated mask $m'$. Here, $|\cdot|$ represents element-wise application of absolute value.

4: Reset the weights of the remaining portion of the network to their values in $\theta^{(0)}$ for 'LTH' experiments and to their values in $\theta^{(3)}$ for 'LTH-Rewind' experiments.

5: Rescale the values of weights based on the sparsity of the network: $\theta^{(0)} \leftarrow \theta^{(0)}/(1 - \frac{\|m'\|_0}{P})$ for 'LTH' experiments and $\theta^{(3)} \leftarrow \theta^{(3)}/(1 - \frac{\|m'\|_0}{P})$ for 'LTH-Rewind' experiments.

6: Let $m = m'$ and repeat steps 2 through 4 until a sufficiently pruned network has been obtained.

---

**SNIP (Single-shot Network Pruning)**: SNIP, introduced by Lee et al. (2018), is another pruning at-initialization technique designed to calculate importance scores based on gradients, preserve weights with the most significant impact on the loss function, and thereby selectively eliminate redundant connections in a data-dependent manner. In the standard implementation of SNIP, all of those operations are executed in a single iteration. To facilitate a fair comparison, we consider a recently introduced iterative variant of SNIP proposed by de Jorge et al. (2020). In our experiments, we utilize the implementation of this iterative SNIP available at `https://github.com/hoonyyhoon/Synflow_SNIP_GraSP`. Also, with 100 iterations performed, weights are gradually pruned in accordance with an exponential schedule as proposed in (de Jorge et al., 2020), ensuring that the final desired level of sparsity is achieved at the 100[th] iteration.

**GraSP (Gradient Signal Preservation)**: GraSP, introduced by Wang et al. (2020), calculate the Hessian-gradient product for each layer to selectively eliminate weights that diminish gradient flow and prunes

weights whose removal results in the least decrease in the gradient norm. We utilize the code available at `https://github.com/hoonyyhoon/Synflow_SNIP_GraSP` for the implementation of GraSP, and adopt the same strategy as implemented by (Wang et al., 2020) for creating the mini-batch used to compute GraSP scores in CIFAR-10 and CIFAR-100 experiments. Specifically, we randomly sample ten examples from each class, aligning with their approach for both CIFAR-10 and CIFAR-100 experiments.

**SynFlow**: SynFlow, proposed by Tanaka et al. (2020), is a data-agnostic pruning at initialization algorithm designed to preserve the total flow of synaptic strengths through the network and achieve Maximum Critical Compression. Our implementation of SynFlow in our experiments follows the strategy outlined in the repository `https://github.com/hoonyyhoon/Synflow_SNIP_GraSP`. To streamline the process, we conduct 100 iterations, mirroring the approach employed by (Tanaka et al., 2020)

---

**Algorithm 3:** Iterative Synaptic Flow Pruning (SynFlow)

---

Randomly initialize a neural network $f_L^s(x; (m \odot \theta^{(0)}))$ where $m = \mathbf{1}$.

**for** $n$ in $N$ **do**

    Replace all parameters $\theta^{(0)}$ with their absolute values $|\theta^{(0)}|$.

    Sample a mini-batch of training data $\mathbb{D}_b = \{(x_i, y_i)\}_{i=1}^b \subset \mathbb{D}$.

    Forward propagate an input of all ones $\mathbf{1}$ through the network $f_L^s(\mathbf{1}; (m \odot |\theta^{(0)}|))$.

    Prune $(1 - S^{n/N})$ fraction of the parameters according to the criterion $\left| \nabla_{\theta^{(0)}} f_L^s(x; m \odot |\theta^{(0)}|) \odot \theta^{(0)} \right|$,

    creating an updated mask $m'$.

    Let $m = m'$.

**end**

Rescale the values of weights based on the sparsity of the network: $\theta^{(0)} \leftarrow \theta^{(0)}/(1 - S)$.

Train the network $f_L^s(x; (m \odot \theta^{(0)}))$ for $E$ epochs with the optimal learning rate to obtain $f_L^s(x; (m \odot \theta^{(E)}))$.

---

**ProsPr**: Alizadeh et al. (2022) present ProsPr, a method that leverages meta-gradients, and incorporates an estimation of the higher-order effects of pruning on the loss function and the optimization trajectory to pinpoint trainable sparse models during initialization. For our comparative analysis, we utilize their implementation at `https://github.com/mil-ad/prospr`.

**GEM-MINER**: GEM-MINER, introduced by Sreenivasan et al. (2022), represents a recent advancement in identifying subnetworks at initialization, revealing winning tickets within initialized subnetworks. It is based on one-shot pruning method, edge-popup (EP) (Ramanujan et al., 2020) for identifying supermasks. The code implementation of GEM-MINER from `https://github.com/ksreenivasan/pruning_is_enough` is used for the experiments.

**Algorithm 4:** GEM-MINER

---

Initialize $f_L^{\mathrm{s}}(x; (m \odot \theta^{(0)}))$, $q$.

**for** epoch $e$ in $E$ **do**

    **for** iteration $n$ in $N$ **do**

        /*** STE ***/

        Train $f_L^{\mathrm{s}}(x; (r(m) \odot \theta^{(0)}) \odot q)$.

        /*** STE ***/

        $m \leftarrow \mathrm{proj}_{[0,1]^P}\, m$.

    **end**

    **if** $\mathrm{mod}(e, V) = 0$ **then**

        $I_1 \leftarrow \{u \mid q_u = 1\}$.

        $m_{\mathrm{sorted}} \leftarrow \mathrm{Sort}\ \{m_u, u \in I_1\}$.

        $m_{\mathrm{bottom}} \leftarrow$ Bottom- 20% of $m_{\mathrm{sorted}}$.

        $q \leftarrow q \odot 1_{m_u \notin m_{\mathrm{bottom}}}$.

    **end**

**end**

Rescale the values of weights based on the sparsity of the network: $\theta^{(0)} \leftarrow \theta^{(0)}/(1 - S)$.

Train the network $f_L^{\mathrm{s}}(x; (q \odot \theta^{(0)}))$ for a total of $E$ epochs using an optimal learning rate, resulting in $f_L^{\mathrm{s}}(x; (q \odot \theta^{(E)}))$.

---

**Continuous Sparsification (CS)**: Continuous Sparsification (CS) (Savarese et al., 2019) is designed to identify sparse subnetworks during the early training phases by approximating $l_0$ regularization through a series of continuous sigmoid functions. We utilize their publicly available code from `https://github.com/lolemacs/continuous-sparsification` to replicate CS results. In our experiments replicating CS, we set the rewinding epoch to the 3rd epoch, employ 3 rounds of training to identify winning tickets, and use hyperparameter values $\lambda = 10^{-8}$, $\beta^{(0)} = 1$, $\beta^{(E)} = 300$. We initialize mask parameters to different constants for varying the sparsity levels across experiments.

---

**Algorithm 5:** Continuous Sparsification

---

**for** round $r$ in $R$ **do**

    Initialize $\beta \leftarrow 1$

    Train $\min\limits_{s, \theta}\ L_{\mathrm{CE}}(x; \sigma(\beta s) \odot \theta) + \lambda \|\sigma(\beta s)\|_1$ for $E$ epochs, while increasing $\beta$ by factor $\mu$ each epoch.

    If round $r = R$, output $f_L^{\mathrm{s}}(x; H(s^{(E)}) \odot \theta^{(3)})$.

    Otherwise, set $s \leftarrow \min\{\beta^{(E)} s^{(E)}, s^{(0)}\}$, $\beta \leftarrow 1$. Here, $\min\{\cdot, \cdot\}$ represent element-wise application of the minimum function to compare corresponding elements of the two vectors.

**end**

Rescale the values of weights based on the sparsity of the network: $\theta^{(3)} \leftarrow \theta^{(3)}/(1 - \frac{\|H(s^{(E)})\|_0}{P})$.

Train the network $f_L^{\mathrm{s}}(x; (H(s^{(E)})) \odot \theta^{(3)}))$ for $E$ epochs with optimal learning rate to obtain $f_L^{\mathrm{s}}(x; (H(s^{(E)})) \odot \theta^{(E)}))$.

---

