# OpenReview forum: "Winning the Lottery Once and For All: Towards Pruning Neural Networks at Initialization"
_TMLR — Rejected by TMLR_

### Review · Reviewer_SiVP · 2024-04-16

**Summary Of Contributions:**

This paper proposes a new method to identify a good sparse subnetwork of an initial model, which can be trained to the original dense model's final test accuracy with very few nonzero weights. The idea is to explicitly train the sparsity masks via straight-through estimator, where the loss is a combination of the cross-entropy loss and several distillation losses. The discovered subnetwork achieves better final test accuracy than its competitors, and also passes the basic sanity checks, such as the randomization of masks.

**Audience:**

Yes

**Broader Impact Concerns:**

Sufficiently addressed.

**Claims And Evidence:**

Yes

**Requested Changes:**

- A comparison with the test accuracy of the dense model, which has been trained with distillation.
- A fairer comparison with other baselines, in terms of the required training cost.
- I recommend adding additional descriptions of the baseline somewhere.
- In p3, there is a typo: "sections 6 and 6 (...)"
- In p3, I am not sure why authors mention Gale et al. I do not think the paper talks about pruning at initialization.
- Please make the distinction between LTS-Rewind and LTS more clearly.

**Strengths And Weaknesses:**

**Strengths**
- The idea of utilizing teacher networks to identify the subnetwork is new to me. It would be interesting to see if one can do similar things for training-free methods like SNIP, where we use the gradient of the distillation loss instead of the vanilla classification loss.
- The accuracy gain of the proposed method is substantial, in my opinion. Especially, they seem to be even better than the dense network.

**Weaknesses**
- The proposed method needs multiple teachers, which can be computationally very expensive to acquire. Thus, it seems rather unfair to me to argue that the proposed method enjoys advantages in terms of the "time saving" (figure 2).
- The method may requires much more RAM to acquire masks than dense training or LTH-like methods. This is because the method requires loading multiple teachers on the memory for taking inferences from the model. Thus, the practical impact of the method seems to be even smaller than the original lottery ticket paper.
- The fact that the method passes the sanity check is not super surprising (still cool), as the method is closer to the LTH than to pruning-at-initialization algorithms.
- I find this paper very difficult to read; I generally think that the text could have been compressed more. The paper has not been very clear to me, especially because the paper does not provide a concise yet informative overview of their method (with much emphasis on how big the paper's contribution is).

---

> ### Author Response · Authors · 2024-06-16
> **Addressing Computational Complexity: Thank You for Your Review.**
>
> Computational Complexity vs. Accuracy: We understand your concern regarding the computational demands of using multiple teacher models. To address it, we are now working on providing a detailed discussion to better elucidate the trade-off between computational complexity and accuracy.
>
> Practical Impact: Regarding the computational demands, we utilize offline distillation, where only the intermediate and final layer responses of the teacher models are stored. This strategy eliminates the need to run inference on the teacher networks during each epoch, significantly reducing the overall computational load during the training of the student model’s masks. We have included ablation studies on teacher models for a LeNet/MNIST setup in the appendix to further investigate the computational complexity specifically associated with teacher models. We plan to extend these studies to include other teacher networks used in various experimental setups.
>
> Typographical and Content Corrections: We will correct the noted typographical errors. Additionally, we will clearly explain the differences between LTS-Rewind and LTS in the main paper.
>
> Thank you for the insightful feedback and support. We truly appreciate your time and effort in reviewing our work. We are committed to addressing your suggestions and will submit the revised paper to the portal as soon as possible.

---

### Review · Reviewer_RWUQ · 2024-05-12

**Summary Of Contributions:**

The paper introduces a new method called Knowledge Distillation-guided Lottery Ticket Search, or KD-LTS. The high level idea is to discover sparse subnetworks (lottery tickets)

**Audience:**

Yes

**Claims And Evidence:**

No

**Requested Changes:**

* (critical) Please tone down wording throughout the paper, simplifying adjectives and tightening the precision of claims wherever possible.
* (strengthen) Please include better captions to figures and tables throughout the paper.
* (strengthen) Please address the following comments.
  - Figure 1 is confusing. There are two arrows that leads into "untrained student model" but no outgoing arrows; why is this block useful then?
  - Figure 2:  "time-saving" is confusing. Also, in general, I would recommend not using spider charts here. Simple bar graphs would suffice.
  - While I respect the stylistic choices made by the authors, I found the intro too flowery and confusing. e.g. (i) the "Key Insight" callout green window on Page 3 has somewhat ambiguous terms (what does "efficiently distills a rich tapestry of information" mean?). (ii) Contribution 1: what does "revitalizing" and "resurrect" mean? (iii) Figure 3 caption: why say "juxtapose" when you likely mean "compare"?  (iv) why say "delve into these open questions" when you just mean "address"? (v) Technical terms like "optimal" are thrown around too loosely, e.g. "orchestrates control of sparsity, playing a pivotal role in striking optimal balance". etc etc.
  - What is "CS" in Table 1? I would recommend having the same list of methods in Figure 3 and Table 1.
  - Figure 3 is very hard to read, especially the color scheme. All the curves basically look superimposed on top of each other. A table might be better.
  - What are the "stops" in Section 4? Why not just say "steps"?
  - Not sure why there is a need to introduce jargon like "mixed integer optimization" since as far as I can tell, the authors immediately relax the l0-regularization term similar to continuous sparsification, and solve it using gradient-based methods.
  - "Figure 9 shows compelling representation of generalization strengths" -- I don't see how Figure 9 shows this. Please use a better way to visualize. In general I find the loss landscape parts unnecessary.
 - Callout on Page 21: Not sure what "randomization ambiguity exclusion is referring to."
  - Conclusion: "This study challenges the conventional wisdom that efficient sparse subnetworks cannot be found at initialization". I certainly don't think this can be called "conventional wisdom".

**Strengths And Weaknesses:**

**Strengths**
* Finding parameter-efficient and performant deep neural networks is a problem of considerable interest to the community.
* Pruning-based methods based on lottery tickets are a promising avenue to achieve this goal.
* The paper contains a very extensive and thorough set of comparative experiments with a large number of published approaches.

**Weaknesses**

* The method is a mashup of (at least) 5-6 different loss terms + assorted algorithmic techniques: Lottery-tickets, knowledge distillation, feature/relation distillation, continuous sparsification, Gem-Miner, and selective freezing. The ablation studies in the appendix study various parameter choices, but one key ablation I would like to see what the impact of turning off the above loss terms and other tricks one-by-one and how that affects overall performance. Otherwise it is hard to deduce the essence of the overall approach.
* The paper is confusingly written, with a ton of unnecessary terms and verbose jargon included throughout. I list several concerns below in the Requested-Changes section.

---

> ### Author Response · Authors · 2024-06-16
> **Enhancing Clarity and Detailing Ablation Studies: Thank You for Your Review.**
>
> Ablation Studies: We value your interest in the specific impact of individual components of our approach. We have conducted ablation studies detailing how different types of information transferred from teacher to student models, specifically in a LeNet/MNIST setup, affect the performance of the identified winning tickets. However, pinpointing which specific knowledge transfer effectively aids in the identification of winning tickets is still an active area of our research. We intend to include detailed findings from these ongoing studies in the section dedicated to future work. This addition will clarify the contributions of various distillation components to the effectiveness of pruned networks.
>
> Simplification and Precision of Language: We agree the need to enhance clarity and are working on this. The forthcoming revision will aim to simplify the language and refine the precision of our claims to make the content more readable.
>
> Figure and Table Captions: We will improve the captions of figures and tables to provide clearer descriptions.
>
> Clarifications and Error Corrections: We are committed to correcting all noted errors, including figure layouts and references.
>
> Randomization Ambiguity Exclusion:  This concept emphasizes that key steps, such as identifying precise parameter values for unpruned weights after mask training, cannot be substituted by randomization procedures—such as selecting random values for unpruned weights—without compromising accuracy. This clarification aims to substantiate the computational complexity of our approach.
>
> Thank you for the insightful feedback and support. We truly appreciate your time and effort in reviewing our work. We are committed to addressing your suggestions and will submit the revised paper to the portal as soon as possible.

---

### Review · Reviewer_ASeh · 2024-05-18

**Summary Of Contributions:**

This paper provides a strongly substantiated case for the possibility of pruning neural networks at initialization, by presenting a novel approach (KD-LTS) for identifying winning tickets (i.e. masks to be applied over model parameters at initialization) based on knowledge distillation. The paper provides a comprehensive evaluation suite of metrics to judge the performance and tradeoffs of different pruning approaches, and extensive empirical results to highlight the success and high value (in terms of accuracy, sparsity, efficiency and stability) of the proposed approach.

**Audience:**

Yes

**Broader Impact Concerns:**

No significant concerns that would require a broader impact statement. Efficiency benefits are well-motivated in the text and an adversarial robustness analysis is given. There is some room to discuss opportunity for an investigation into risks of pruning increasing bias.

**Claims And Evidence:**

Yes

**Requested Changes:**

### Substantial changes:
* Move much of the related work to the appendix - particularly section 2, as well as trimming down sections 1 and 4.
* Cut down repetition of the high level success of KD-LTS in the main text to provide more concrete evidence and specific details in the body of the paper.
* Include information about the empirical setup - particularly the teacher models (including architecture, parameter count, number of teachers, training procedure, training costs) and how the KD-LTS algorithm is configured to trade off different objectives - in the main body of the text. Without this information it is difficult to understand the method in full and grasp the computational efficiency vs accuracy tradeoff of this approach in relation to other methods.
* Point out the limitations of the approach and where the empirical analyses are lacking. What cases did you not consider? Where are there gaps and opportunities for future work? Ideally this should be mentioned in the main text, rather than just the appendix.

### Minor changes:
* If possible, moving figures to better align with the text would help with readability. Additionally, making the order in which the figures are placed and where they are first referenced in the text consistent would make it easier to follow.
* Nit: incorrect reference on page 3 “…sensitivity and stability analysis, readers are encouraged to refer to Sections 6 and 6, respectively.”

**Strengths And Weaknesses:**

### Strengths

The paper is well written and presented in a way that is easy to follow, with key findings highlighted. The motivation behind pruning at initialization is clear, and the background provided is quite comprehensive so readers unfamiliar with the landscape can understand the space. The impact of this work applies not only to the pruning field but also provides key insights into the research area of knowledge distillation: whereas current KD approaches focus on training a small dense student model, there might be promise in applying these techniques to training a large pruned student model.

This paper presents a multi-stage algorithm that composes existing techniques, and provides extensive empirical analyses to validate its promise. Experimental results compare KD-LTS to a number of existing approaches, evaluating the resulting subnetworks according to their accuracy, sparsity, efficiency, stability, sensitivity, transferability and robustness. The comprehensive evaluation suite and definitions of relevant metrics that this work provides is a key contribution to the sparsity field to better analyze and assess the tradeoffs of various methods.

There is some interesting analysis demonstrating how the approach can allow for trading off different objectives and as a result yield specialized characteristics. This is an interesting direction of study to strategically apply pruning for targeting specific traits.


### Weaknesses

The paper is quite long and many of the key findings and contributions (even those called out in the introduction) are relegated to the appendix. For example, the paper omits from the main text how teacher models are trained, even just high level details around the scale of the models and numbers of teachers needed. This information is critical to grounding the approach and the results presented in a concrete rationale of the compute requirements and tradeoffs for using KD-LTS. The discussion of computational efficiency is not in the main text even though it is a primary motivation of the work.

The current main text and appendices could be condensed and focused to give room for discussion and analysis of the results, providing a more compelling, crisp and complete argument in the main body of the paper. Much of the related work could be moved to the appendix, as could many of the plots that take up significant space in the main text and are not fully discussed or referenced. Throughout the text (both in the main body and appendix), prior work could be referenced but not fully explained so as to tighten the paper to just focus on and clarify the contributions of this work. There is so much context provided that it is hard to understand the specifics of this paper or what is important to focus on.

Overall, this paper would benefit from being significantly pruned to better highlight its contributions, insights and key results!

---

> ### Author Response · Authors · 2024-06-16
> **Enhancing Focus and Detail in Our Manuscript: Thank You for Your Review.**
>
> Refinement and Conciseness: We appreciate the suggestion to enhance the readability and focus of our manuscript. To this end, we are working on moving discussions on "Assessment of the Training Landscape", "Examination across the Optimization Landscape", and "Validating the Quality of Winning Tickets" to the appendix. The main text will now concentrate exclusively on the critical and novel aspects of the KD-LTS method.
>
> Detailed Empirical Setup: We recognize the importance of providing clear and comprehensive details about teacher models. The revision will now include detailed descriptions in the main text regarding the teacher models employed. For instance, we utilized three ResNet-20 teacher models to prune a ResNet-20 student model. By employing offline distillation, we stored only the intermediate and final layer responses corresponding to each data sample, eliminating the need to run inference on the teacher networks every epoch. This approach significantly reduces computational complexity while effectively training the student model's masks. Ablation studies on reducing the computational complexity around teacher models for a LeNet/MNIST setup have been included in the appendix. Information regarding the training and optimization settings of the teacher models is also provided in the appendix for clarity.
>
> Future Works and Limitations: We are revising and enhancing the section on future works and limitations within the main text, moving discussions from the appendix to ensure they receive proper emphasis. This section will detail our ongoing efforts to utilize pre-trained teacher models that not only have different architectures but are also trained on varied datasets, and to utilize distinctly different training regimes compared to the student models. These investigations are aimed at further validating the efficiency of the KD-LTS framework.
>
> Thank you for the insightful feedback and support. We truly appreciate your time and effort in reviewing our work. We are committed to addressing your suggestions and will submit the revised paper to the portal as soon as possible.

---

### Author Response · Authors · 2024-06-16
**Response to All Reviewers: Thank You for Your Feedback.**

Thank you all so much for the insightful reviews. Your recommendations have been invaluable in guiding us toward improving our paper. We appreciate your time and effort in providing detailed feedback.

We are diligently working on revising the paper according to your valuable suggestions. As you have delightfully put it, our "paper needs to be pruned," and we are indeed making it more concise and focused.

Highlighting Key Contributions: We will make the key contributions of the paper clearer, which are: 1) Introduction of Knowledge Distillation Aided Pruning at Initialization, which offers all-encompassing performance improvements in terms of accuracy, sparsity, robustness, stability, fidelity, and transferability, and 2) New Regime of Knowledge Distillation, wherein large networks are pruned for deployment and transferring special characteristics beyond test accuracy using specialized teacher models.

Reorganization of the Paper: We will reorganize the main text and appendix as suggested. We will move much of the related work to the appendix and address future work and limitations explicitly in the main text.

Teacher Network Complexity: We will include detailed information about the empirical setup, particularly the teacher models, in the main body of the text. We have addressed teacher network complexity in the appendix for a LeNet/MNIST setting, including the behavior of KD-LTS pruned student networks trained using teacher networks trained to different stages and using different temperatures, the number of teacher models, different information transfer techniques, and using a reduced dataset with their corresponding implications. To address the accuracy vs. computational complexity trade-off, we will extend this discussion to include the complexity of teacher models for the networks and datasets used in the main paper. Since we use offline distillation in the main paper for our experiments, the teacher networks need to be fed data to observe intermediate and final layer outputs once, which can be stored for all further training epochs.

Clarifying Contributions and Refining Language: We will work on making our contributions clearer and refining the language to ensure precision and simplicity. Additionally, we will incorporate minor revisions and correct mistakes as mentioned. We will move figures to better align with the text, ensure consistency in their placement, and provide proper captioning.

Randomization ambiguity exclusion: We mention "randomization ambiguity exclusion" to emphasize that steps in the KD-LTS process cannot be replaced by a randomization step, such as random choice of unpruned weights, which would call into question the existing step's computational complexity.

Once again, we thank you for your valuable feedback and the opportunity to improve our paper. We are committed to addressing all your suggestions and will upload the revised paper as soon as possible.

---

### Decision · Action_Editor_DvNV · 2024-06-27

**Recommendation:** Reject

**Comment:**

The paper studied an important problem that could potentially be interesting to the audience in the ML community. The reviewers have pointed out various presentation and technical concerns about the current version of the paper. For example, the paper should be re-organized to highlight the descriptions of the main setting and emphasize the key contributions in the main text, e.g., the training process of teacher models and the computational efficiency should be described in the main text. The paper should be precise about technical terms, e.g., the word "optimal" should be used in a careful manner with clear indication of optimality. The comparison with other baselines should be under the same computational complexity to be fair. There are also other important comments in the detailed reviews.

Unfortunately, the authors did not make the corresponding revision in the paper following the reviewers' suggestions.

**Audience:**

Yes.

**Claims And Evidence:**

The paper contains some claims that are not precise, as pointed out by Reviewers RWUQ and SiVP.

**Resubmission Of Major Revision:**

The authors may consider submitting a major revision at a later time.